# Transformer-based spatial–temporal detection of apoptotic cell death in live-cell imaging

**Alain Pulfer[1,2†], Diego Ulisse Pizzagalli[1,3†], Paolo Armando Gagliardi[4†], Lucien Hinderling[4], Paul Lopez[5], Romaniya Zayats[5], Pau Carrillo-Barberà[1,6], Paola Antonello[1,4], Miguel Palomino-Segura[7,8], Benjamin Grädel[4], Mariaclaudia Nicolai[3], Alessandro Giusti[9], Marcus Thelen[1], Luca Maria Gambardella[9], Thomas T Murooka[5], Olivier Pertz[4], Rolf Krause[3], Santiago Fernandez Gonzalez[1]***

[1]Institute for Research in Biomedicine, Faculty of Biomedical Sciences, USI, Lugano, Switzerland; [2]Department of Information Technology and Electrical Engineering, ETH Zurich, Zürich, Switzerland; [3]Euler Institute, USI, Lugano, Switzerland; [4]Institute of Cell Biology, University of Bern, Bern, Switzerland; [5]University of Manitoba, Winnipeg, Canada; [6]Instituto de Biotecnología y Biomedicina (BioTecMed), Universitat de València, Valencia, Spain; [7]Immunophysiology Research Group, Physiology Department, Faculty of Sciences, University of Extremadura, Badajoz, Spain; [8]Immunophysiology Research Group, Instituto Universitario de Investigación Biosanitaria de Extremadura (INUBE), Badajoz, Spain; [9]Dalle Molle Institute for Artificial Intelligence, IDSIA, Lugano, Switzerland

*For correspondence:
santiago.gonzalez@irb.usi.ch

†These authors contributed equally to this work

**Competing interest:** The authors declare that no competing interests exist.

## eLife assessment

This **valuable** study advances our understanding of spatial–temporal cell dynamics both in vivo and in vitro. The authors provide **solid** evidence for their innovative deep learning-based apoptosis detection system, ADeS, which utilizes the principle of activity recognition. This work will be of broad interest to cell biologists and neuroscientists.

**Abstract** Intravital microscopy has revolutionized live-cell imaging by allowing the study of spatial–temporal cell dynamics in living animals. However, the complexity of the data generated by this technology has limited the development of effective computational tools to identify and quantify cell processes. Amongst them, apoptosis is a crucial form of regulated cell death involved in tissue homeostasis and host defense. Live-cell imaging enabled the study of apoptosis at the cellular level, enhancing our understanding of its spatial–temporal regulation. However, at present, no computational method can deliver robust detection of apoptosis in microscopy timelapses. To overcome this limitation, we developed ADeS, a deep learning-based apoptosis detection system that employs the principle of activity recognition. We trained ADeS on extensive datasets containing more than 10,000 apoptotic instances collected both in vitro and in vivo, achieving a classification accuracy above 98% and outperforming state-of-the-art solutions. ADeS is the first method capable of detecting the location and duration of multiple apoptotic events in full microscopy timelapses, surpassing human performance in the same task. We demonstrated the effectiveness and robustness of ADeS across various imaging modalities, cell types, and staining techniques. Finally, we employed ADeS to quantify cell survival in vitro and tissue damage in mice, demonstrating its potential application in toxicity assays, treatment evaluation, and inflammatory dynamics. Our findings suggest that

ADeS is a valuable tool for the accurate detection and quantification of apoptosis in live-cell imaging and, in particular, intravital microscopy data, providing insights into the complex spatial–temporal regulation of this process.

## Introduction

In the last two decades, intravital microscopy (IVM) has revolutionized live-cell imaging by enabling microscopy acquisitions in situ across different organs, making it one of the most accurate models to describe cellular activities within a living host (*Sumen et al., 2004*). In particular, multiphoton intravital microscopy (MP-IVM) generates in-depth 3D data that encompass multiple channels for up to several hours of acquisition (x,y,z+t) (*Helmchen and Denk, 2005*; *Rocheleau and Piston, 2003*; *Secklehner et al., 2017*), thus providing unprecedented insights into cellular dynamics and interactions (*Pizzagalli et al., 2019*). The resulting MP-IVM data stream is a complex and invaluable source of information, contributing to enhance our understanding of several fundamental processes (*Beltman et al., 2009*; *Sumen et al., 2004*).

Apoptosis is a form of regulated cell death (*D'Arcy, 2019*; *Tang et al., 2019*) that plays a crucial role in several biological functions, including tissue homeostasis, host protection, and immune response (*Opferman, 2008*). This process relies on the proteolytic activation of caspase-3-like effectors (*Shalini et al., 2015*), which yields successive morphological changes that include cell shrinkage, chromatin condensation, DNA fragmentation, membrane blebbing (*Elmore, 2007*; *Galluzzi et al., 2018*; *Saraste and Pulkki, 2000*), and finally, apoptotic bodies formation (*Coleman et al., 2001*). Due to its crucial role, dysregulations of apoptosis can lead to severe pathological conditions, including chronic inflammatory diseases and cancer (*Fesik, 2005*; *Hotchkiss and Nicholson, 2006*). Consequently, precise tools to identify and quantify apoptosis in different tissues are pivotal to gain insights on this mechanism and its implications at the organism level.

Traditional techniques to quantify apoptosis rely on cellular staining on fixed cultures and tissues (*Atale et al., 2014*; *Kyrylkova et al., 2012*; *Loo, 2011*; *Sun et al., 2008*; *Vermes et al., 1995*) or flow cytometry (*Darzynkiewicz et al., 2008*; *Vermes et al., 1995*). However, these methods do not allow the temporal characterization of the apoptotic process. Moreover, they potentially introduce artifacts caused by sample fixation (*Schnell et al., 2012*). Live-cell imaging can overcome these limitations by unraveling the dynamic aspects of apoptosis with the aid of fluorescent reporters, such as Annexin staining (*Atale et al., 2014*) or the activation of caspases (*Takemoto et al., 2003*). However, the use of fluorescent probes in vivo could potentially interfere with physiological functions or lead to cell toxicity (*Jensen, 2012*). For these reasons, probe-free detection of apoptosis represents a critical advancement in the field of cell death.

Computational methods could address this need by automatically detecting individual apoptotic cells with high spatial and temporal accuracy. In this matter, deep learning (DL) and activity recognition (AR) could provide a playground for the classification and detection of apoptosis based on morphological features (*Poppe, 2010*). Accordingly, recent studies showed promising results regarding the classification of static frames (*Kranich et al., 2020*; *Verduijn et al., 2021*) or timelapses (*Mobiny et al., 2020*) portraying single apoptotic cells. However, none of the available methods can be applied for the detection of apoptosis in microscopy movies depicting multiple cells. Therefore, we developed ADeS, a novel apoptosis detection system that employs a transformer DL architecture and computes the location and duration of multiple apoptotic events in live-cell imaging. Here, we show that our architecture outperforms state-of-the-art DL techniques and efficiently detects apoptotic events in a broad range of imaging modalities, cellular staining, and cell types.

## Results

### An in vitro and in vivo live-cell imaging data

Curated and high-quality datasets containing numerous instances of training samples are critical for developing data-hungry methods such as supervised DL algorithms (*Adadi, 2021*). To this end, we generated two distinct datasets encompassing epithelial cells (in vitro) and leukocytes (in vivo) undergoing apoptotic cell death. In addition, the two datasets include different imaging modalities (confocal and intravital two-photon), biological models, and training-set dimensionalities. A

meaningful difference between the datasets pertains to the staining methods and the morphological hallmarks, which define the apoptotic process in both models. In the in vitro model, the expression of nuclear markers allowed us to observe apoptotic features such as chromatin condensation and nuclear shrinkage (*Saraste and Pulkki, 2000*), whereas in the in vivo model, cytoplasmic and membrane staining highlighted morphological changes such as membrane blebbing and the formation of apoptotic bodies (*Saraste and Pulkki, 2000*). Accordingly, we have manually annotated these datasets based on the presence of the specific hallmarks, ensuring that each dataset includes two class labels depicting either apoptotic or nonapoptotic cells. These two datasets constitute the first step toward creating, testing, and validating our proposed apoptosis detection routine.

To generate the in vitro dataset, we used epithelial cells because, among the human tissues, they have the highest cellular turnover driven by apoptosis (*van der Flier and Clevers, 2009*). Nevertheless, from the bioimaging perspective, the epithelium is a densely packed tissue with almost no extracellular matrix, making it extremely challenging to analyze. As such, in epithelial research, there is a pressing need for computational tools to identify apoptotic events automatically. To this end, we imaged and annotated the human mammary epithelial cells expressing a nuclear fluorescent marker (*Figure 1A*), obtaining 13,120 apoptotic nuclei and 301,630 nonapoptotic nuclei image sequences (*Figure 1B and C*, *Figure 1—figure supplement 1A*). Nuclear shrinkage and chromatin condensation, two of the most prototypical hallmarks of apoptosis (*Figure 1C*), formed our criteria for manual annotation. We confirmed that nonapoptotic nuclei had constant area and chromatin density from the generated timelapses. In contrast, apoptotic nuclei underwent a decrease in area and an increase in chromatin condensation (*Figure 1D*). The resulting dataset captured the heterogeneity of apoptotic cells in epithelial tissue, including early nuclear fragmentation, a rapid shift along the x and y axes, and extrusion through the z dimension (*Figure 1—figure supplement 1B and C*). Moreover, our dataset incorporates the typical difficulties of automatically annotating apoptotic events from live microscopy of a densely packed tissue (*Figure 1—figure supplement 1D*) with the accumulation of apoptotic bodies (*Figure 1—figure supplement 1E*) and across multiple microscope hardware settings (*Figure 1—figure supplement 1F*).

To generate an in vivo dataset, we focused on polymorphonucleated leukocytes (neutrophils and eosinophils) that expressed a fluorescent marker. In these early immune responders, apoptosis is a crucial process that orchestrates their disposal, consequently determining the duration of the inflammation (*Fox et al., 2010*). To acquire instances of apoptotic leukocytes, we performed MP-IVM in anesthetized mice by surgically exposing either the spleen or the popliteal lymph node (*Figure 1E and F*). The resulting timelapses (*Figure 1G*) provided 3D imaging data encompassing consecutive multifocal planes (3D) and multiple imaging channels. Then, from the generated MP-IVM movies, we generated cropped sequences of fixed size that tracked apoptotic cells for the duration of their morphological changes (59 × 59 pixels + time; *Figure 1H and I*). This procedure was applied to 30 MP-IVM movies, generating 120 apoptotic sequences (*Figure 1—figure supplement 1G*). Furthermore, we annotated random instances of nonapoptotic events, generating 535 cropped samples. To characterize the heterogeneity of the movies, we manually quantified the cell number per field of view (87 ± 76), the shortest distance between cells (21.2 μM ± 15.4), and the signal-to-noise ratio (SNR) (8.9 ± 3.6; *Figure 1—figure supplement 1H–J*). We assumed that the morphological changes associated with apoptosis occur within defined time windows for detection purposes. Hence, we estimated the median duration of the morphological changes corresponding to eight frames (*Figure 1—figure supplement 1K and L,* respectively). In addition, to classify apoptotic cells within defined spatial regions, we considered them to be nonmotile. This assumption was confirmed when we found that apoptotic cells, despite having a longer track length due to passive transport, exhibited a speed that was not significantly different from those of arrested cells (*Figure 1—figure supplement 1M*).

## ADeS: A pipeline for apoptosis detection

Detecting apoptosis in live-cell imaging is a two-step process involving the correct detection of apoptotic cells in the movies (x,y) and the correct estimation of the apoptotic duration (*t*). To fulfill these requirements, we designed ADeS as a set of independent modules assigned to distinct computational tasks (*Figure 2*). As an input, ADeS receives a 2D representation of the microscopy acquisitions (*Figure 2A*) obtained from the normalization of 2D raw data or the maximum projection of 3D data (*Shi, 2015*). This processing step ensures the standardization of the input, which might differ

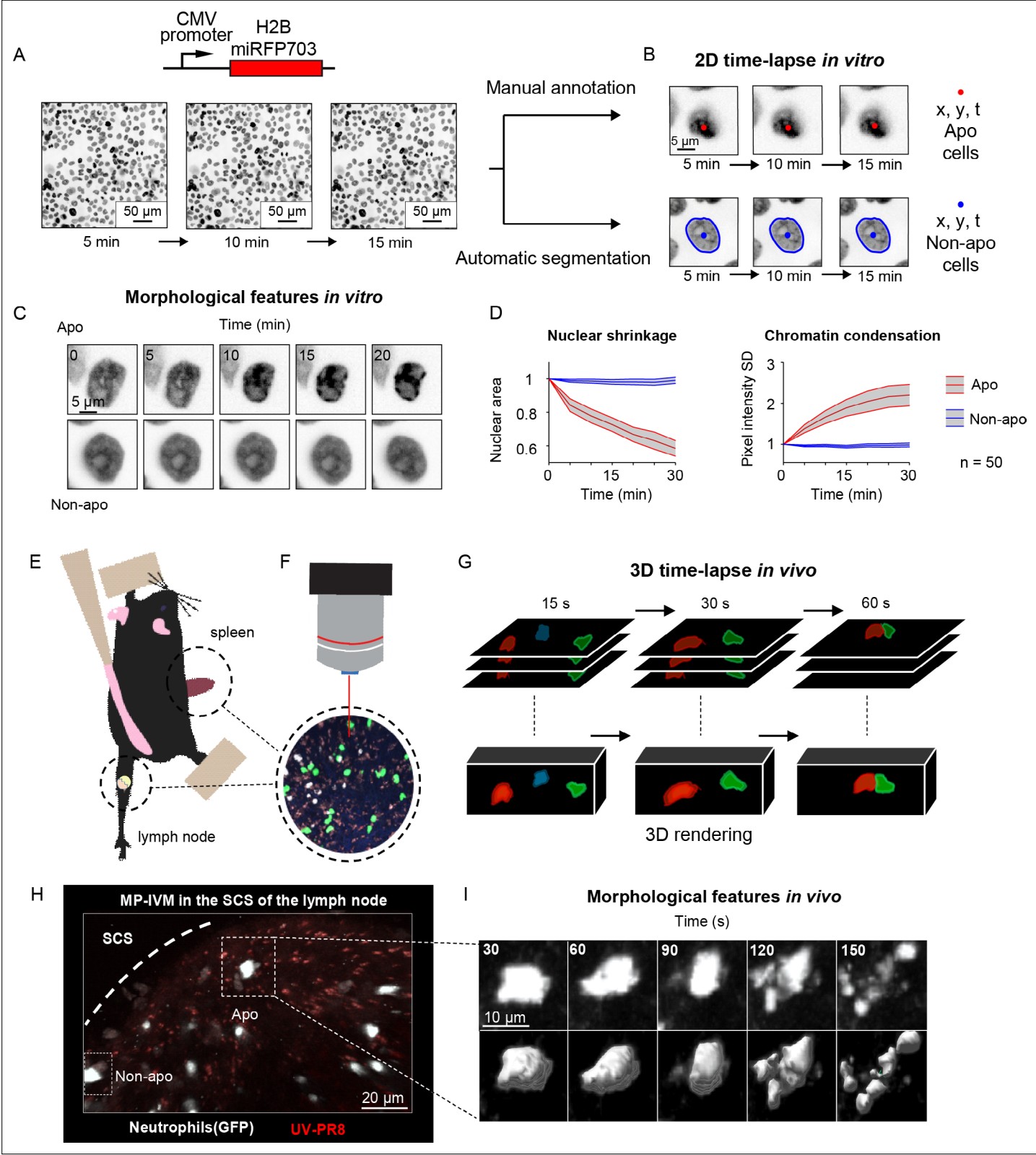

**Figure 1.** Generation of in vitro and in vivo live-cell imaging data. (**A**) Micrographs depicting mammary epithelial MCF10A cells transduced with H2B-miRFP703 marker and grown to form a confluent monolayer. The monolayer was acquired with a fluorescence microscope for several hours with 1, 2, or 5 min time resolution. (**B**) The centroid (x, y) and the time (t) of apoptotic events were annotated manually based on morphological features associated with apoptosis. Nonapoptotic cells were identified by automatic segmentation of nuclei. (**C**) Image timelapses showing a prototypical apoptotic event

*Figure 1 continued on next page*

*Figure 1 continued*

(upper panels), with nuclear shrinkage and chromatin condensation, and a nonapoptotic event (bottom panels). (**D**) Charts showing the quantification of nuclear size (left) and the standard deviation (SD) of the nuclear pixel intensity (right) of apoptotic and nonapoptotic cells (n = 50). Central darker lines represent the mean, and gray shades bordered by light-colored lines represent the standard deviation. Nuclear area over time expressed as the ratio between areas at Tn and T0. (**E**) Simplified drawing showing the surgical setup for lymph node and spleen. (**F, G**) Organs are subsequently imaged with intravital two-photon microscopy (IV-2PM, **F**), generating 3D timelapses (**G**). (**H**) Representative IV-2PM micrograph and (**I**) selected crops showing GFP-expressing neutrophils (white) undergoing apoptosis. The apoptosis sequence is depicted by raw intensity signal (upper panels) and 3D surface reconstruction (bottom panels).

The online version of this article includes the following figure supplement(s) for figure 1:

**Figure supplement 1.** Generation of in vitro and in vivo microscopy data.

in bit depth or acquisition volume. After that, we employ a selective search algorithm (*Girshick, 2015*; *Uijlings et al., 2013*) to compute regions of interest (ROIs) that might contain apoptotic cells (*Figure 2B*). For each ROI at time (*t*), ADeS extracts a temporal sequence of *n* frames ranging from *t − n/2* to *t + n/2* (*Figure 2C*). The resulting ROI sequence is standardized in length and passed to a DL classifier (*Figure 3*), which determines whether it is apoptotic or nonapoptotic. Finally, each apoptotic sequence is depicted as a set of bounding boxes and associated probabilities (*Figure 2D*) generated from the predicted trajectories (x, y, t, ID; *Figure 2E*). From this readout, ADeS can generate a heatmap representing the likelihood of apoptotic events throughout a movie (*Figure 2F*, left), together with a cumulative sum of the predicted cell deaths (*Figure 2F*, right).

For the classification of apoptotic sequences, we proposed a Conv-Transformer architecture (*Figure 3*). In the proposed architecture, a convolutional module extracts the spatial features of the apoptotic cells, whereas attention-based blocks evaluate the temporal relationship between consecutive frames.

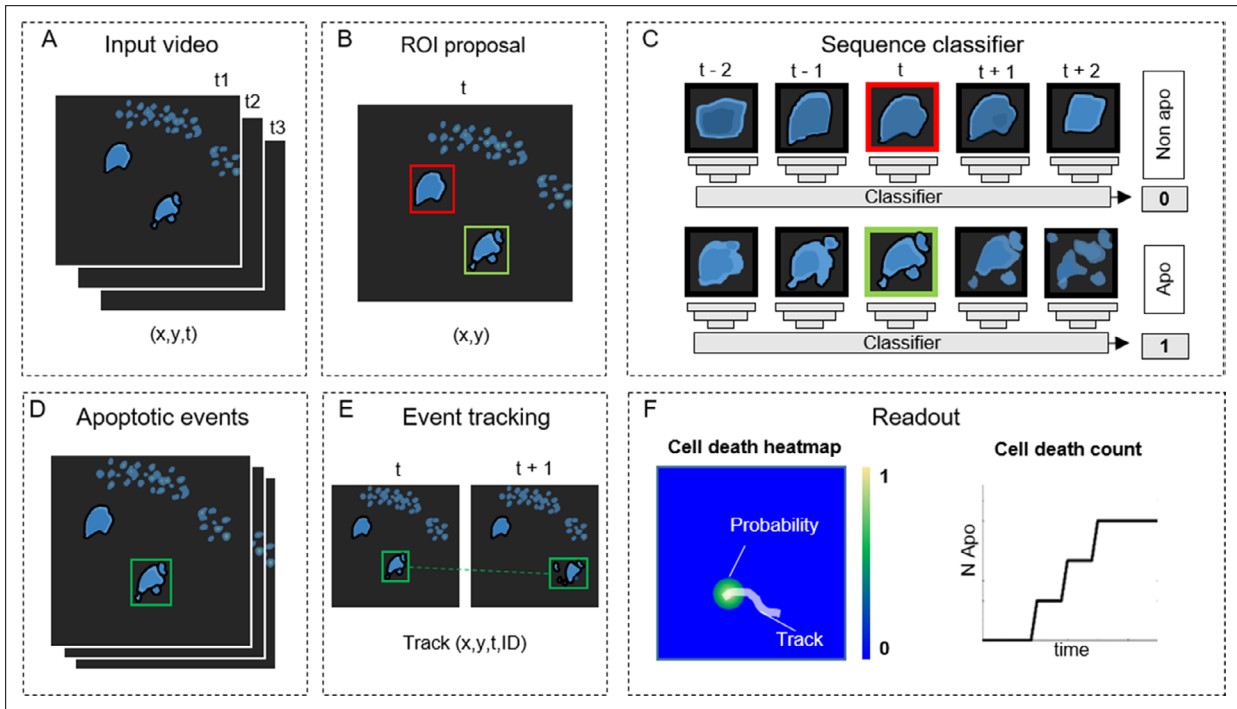

**Figure 2.** ADeS: a pipeline for apoptosis detection. (**A**) ADeS input consists of single-channel 2D microscopy videos (x,y,t) (**B**) Each video frame is preprocessed to compute the candidate regions of interest (ROI) with a selective search algorithm. (**C**) Given the coordinates of the ROI at time *t*, ADeS extracts a series of snapshots ranging from *t − n* to *t + n*. A deep learning network classifies the sequence either as nonapoptotic (0) or apoptotic (1). (**D**) The predicted apoptotic events are labeled at each frame by a set of bounding boxes that (**E**) are successively linked in time with a tracking algorithm based on Euclidean distance. (**F**) The readout of ADeS consists of bounding boxes and associated probabilities, which can generate a probability map of apoptotic events over the course of the video (left) as well as providing the number of apoptotic events over time (right).

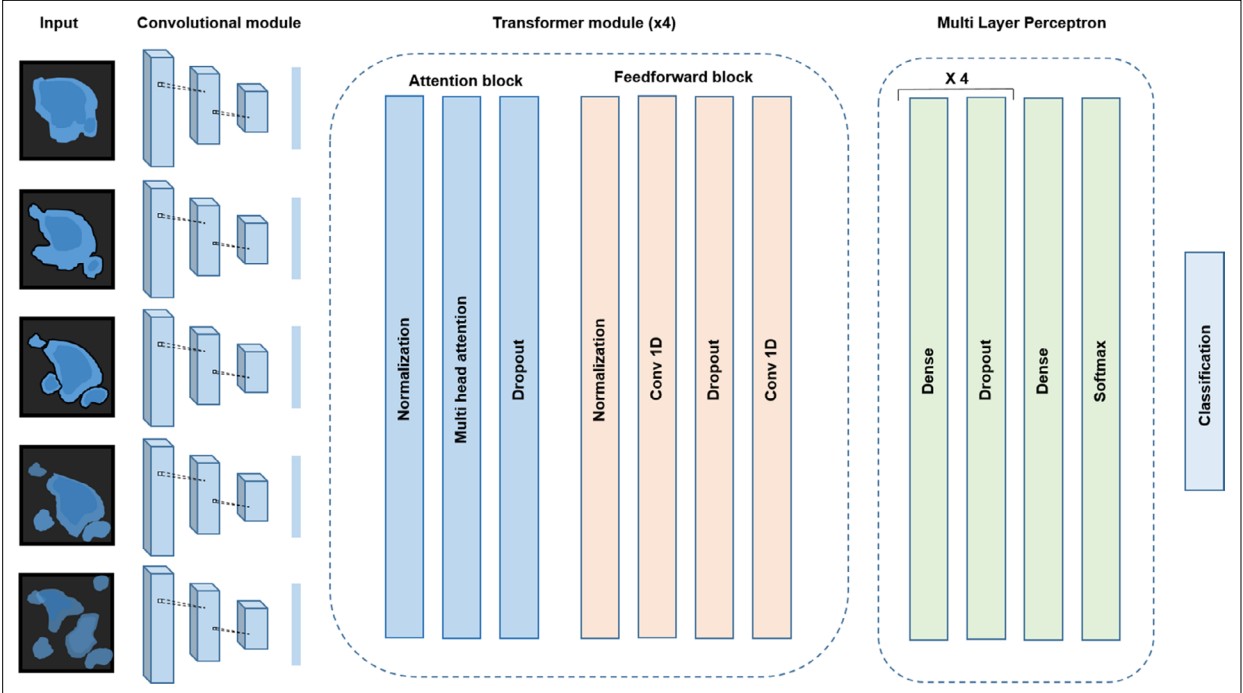

**Figure 3.** Conv-Transformer architecture at the core of ADeS. Abstracted representation of the proposed Conv-Transformer classifier. The input sequence of frames is processed with warped convolutional layers, which extract the features of the images. The extracted features are passed into the four transformer modules, composed of attention and feedforward blocks. Finally, a multilayer perceptron enables classification between apoptotic and non-apoptotic sequences.

## Training and deployment in vitro

As previously described, ADeS is a multiple-block pipeline, and its application and validation to detect apoptotic cells in live-cell imaging follow two main steps: (1) the training of the DL classifier with a target dataset and (2) its deployment on live-cell imaging acquisitions. As opposed to in vivo acquisitions, in vitro timelapses are more homogeneous in their content and quality, thus representing the first dataset in order of complexity for the training of ADeS (*Figure 4*). For this reason, we formulated the learning problem as a binary classification task that assigned nonapoptotic sequences to the class label 0 and apoptotic sequences to the class label 1 (*Figure 4—figure supplement 1A*). The class label 0 included instances of healthy nuclei and nuclei undergoing mitotic division (which can resemble apoptotic events).

Successively, to validate the proposed Conv-Transformer architecture for apoptosis classification, we compared it with the performances of a convolutional neural network (CNN), a 3DCNN, and a convolutional long-short term memory (Conv-LSTM) network. To this end, the four models were trained on a dataset containing 13.120 apoptotic and 13.120 nonapoptotic events using a 0.12 validation split (*Table 1*). Results show that the frame accuracy of the CNN is low, possibly due to morphological heterogeneity over consecutive frames, and therefore unsuitable for the task. By contrast, the 3DCNN and the Conv-LSTM displayed high-sequence accuracy, F1 score, and area under the curve (AUC), confirming that the temporal information within frames is pivotal to correctly classifying image sequences containing apoptotic cells. Nonetheless, the proposed Conv-Transformer outperformed both the 3DCNN and the Conv-LSTM, establishing itself as the final DL architecture at the core of ADeS.

Successively, we deployed a preliminary trained network on control movies without apoptotic events to collect false positives that we used to populate the class label 0, thus ensuring a systematic decrease in the misclassification rate (*Figure 4—figure supplement 1B*). Using the latter generated dataset, we trained the Conv-Transformer for 100 epochs using an unbalanced training set with a 1:10 ratio of apoptotic to nonapoptotic cells (*Figure 4A*). After deploying the trained model on 1000 testing samples, the confusion matrix (*Figure 4B*) displayed a scant misclassification rate (2.68%),

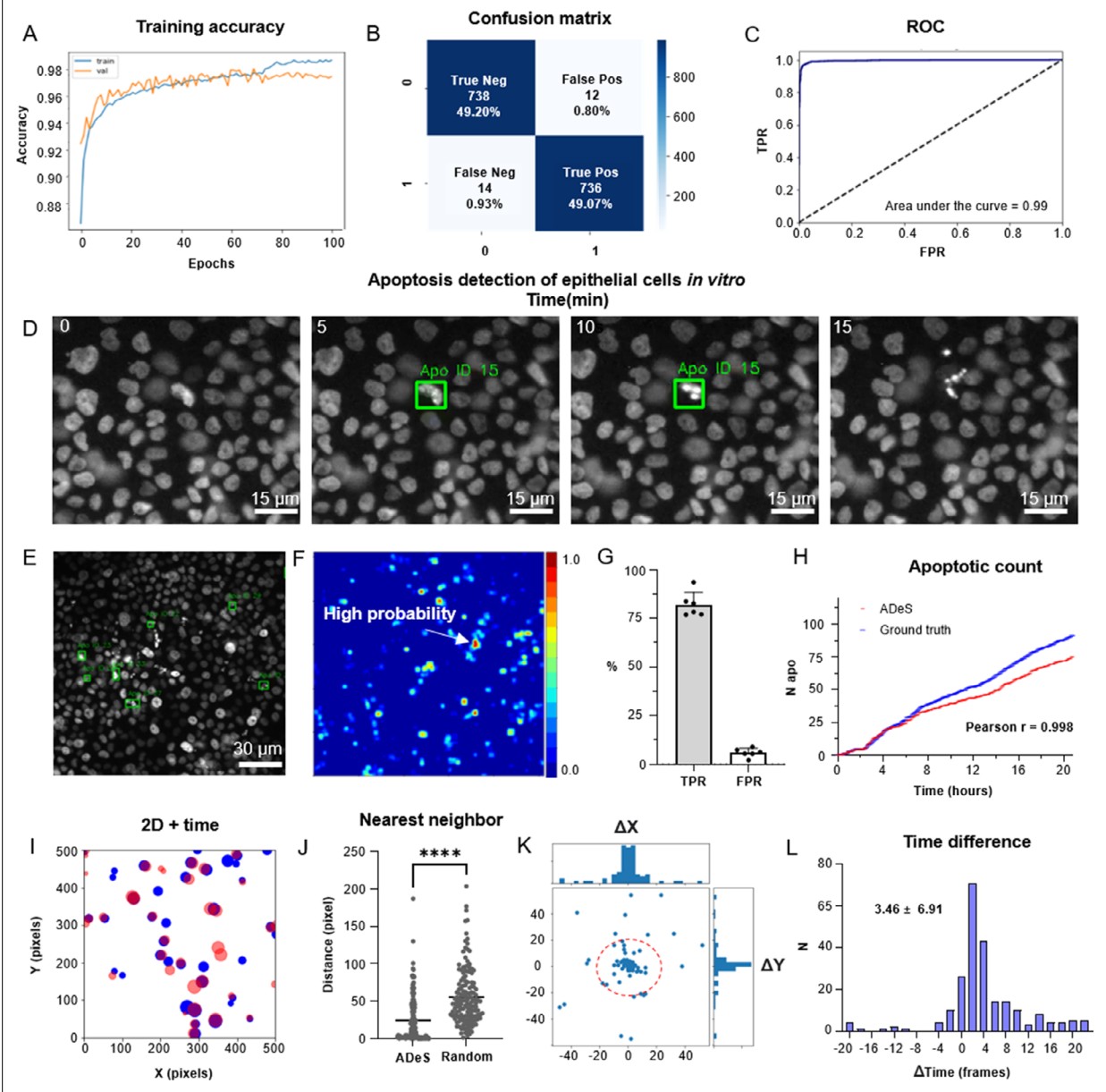

**Figure 4.** Training and performance in vitro. (**A**) Confusion matrix of the trained model at a decision-making threshold of 0.5. (**B**) Receiver-operating characteristic displaying the false positive rate (FPR) (specificity) corresponding to each true positive rate (TPR) (sensitivity). (**C**). Training accuracy of the final model after 100 epochs of training. (**D**) Representative example of apoptosis detection in a timelapse acquired in vitro (five replicates). (**E**) Multiple detection of nuclei undergoing apoptosis displays high sensitivity in densely packed field of views. (**F**) Heatmap representation depicting all apoptotic events in a movie and the respective probabilities. (**G**) Bar plots showing the TPR and FPR of ADeS applied to five testing movies, each one depicting an average of 98 apoptosis. (**H**) Time course showing the cumulative sum of ground-truth apoptosis (blue) and correct predictions (red). (**I**) 2D visualization of spatial–temporal coordinates of ground-truth (blue) and predicted apoptosis (red). In the 2D representation, the radius of the circles maps the temporal coordinates of the event. (**J**) Pixel distance between ADeS predictions and the nearest neighbor (NN) of the ground truth (left) in comparison with the NN distance obtained from a random distribution (right). The plot depicts all predictions of ADeS, including true positives and false positives. (**K**) Scatterplot of the spatial distance between ground truth and true positives of ADeS. Ground-truth points are centered on the X = 0 and Y = 0 coordinates. (**L**) Distribution of the temporal distance (frames) of the correct predictions from the respective ground-truth NN. Statistical comparison was performed with Mann–Whitney test. Columns and error bars represent the mean and standard deviation, respectively. Statistical significance is expressed as *p≤0.05, **p≤0.01, ***p≤0.001, ****p≤0.0001.

The online version of this article includes the following figure supplement(s) for figure 4:

**Figure supplement 1.** Training and performance in vitro.

**Figure supplement 2.** Effect of noise on ADeS performance.

**Table 1.** Comparison of deep learning architectures for apoptosis classification.

Comparative table reporting accuracy, F1, and AUC metrics for a CNN, 3DCNN, Conv-LSTM, and Conv-Transformer. The classification accuracy is reported for static frames or image sequences. The last column shows which cell death study employed the same baseline architecture displayed in the table.

| Classifier architecture | Frame accuracy | Sequence accuracy | F1 | AUC | Study |
|---|---|---|---|---|---|
| CNN | 74% ± 1.3 | NA | 0.77 | 0.779 | *La Greca et al., 2021*; *Verduijn et al., 2021* |
| 3DCNN | NA | 91.22 % ± 0.15 | 0.91 | 0.924 | - |
| Conv-LSTM | NA | 97.42% ± 0.09 | 0.97 | 0.994 | *Kabir et al., 2022*; *Mobiny et al., 2020* |
| Conv-Transformer | NA | 98.27% ± 0.25 | 0.98 | 0.997 | Our |

CNN, convolutional neural network; NA, nonapplicable.

similarly distributed between false positives (1.04%) and false negatives (1.64%). Accordingly, the receiver-operating characteristic (ROC) of the model skewed to the left (AUC = 0.99, *Figure 4C*). This skew indicates a highly favorable tradeoff between the true positive rate (TPR) and false positive rate (FPR), which the overall predictive accuracy of 97.32% previously suggested (*Figure 4B*). Altogether, these metrics suggest an unprecedented accuracy of the DL model in the classification of apoptotic and nonapoptotic sequences. However, they only reflect the theoretical performances of the classifier applied to cropped sequences depicting a single cell at a time.

To validate ADeS on full-length microscopy acquisitions, we deployed it on six testing movies that were not part of the training set. Each testing movie had been annotated manually and contained a variable number of ground-truth apoptosis (98 ± 21) and a comparable cell density (1705 ± 124). Moreover, all movies had identical magnification (20×), duration (21 hr), and sampling rate (5 min). In order to test ADeS on these movies, we adopted an unbiased approach and we did not hard-tune the hyperparameters of the model (see 'Materials and methods'), specifying only a stringent confidence threshold (0.995) and a temporal window based on the average duration of the nuclear hallmarks (nine frames). As a result, ADeS could predict the location and timing of the apoptotic nuclei (*Figure 4D*, *Video 1*), enabling the detection of multiple apoptoses in a densely packed field of view (*Figure 4E and F*). To quantify these performances, we compared the prediction of ADeS to the annotated ground truths (x,y,t). By doing this, we found that the average TPR, or sensitivity, was 82.01% (ranging from 77 to 92%), while the average FPR was 5.95% (*Figure 4G*). The undetected apoptotic events were likely a consequence of the heterogeneity of nuclear fragmentation, which can vastly differ in signal intensity, size, focal plane, and duration (*Figure 1—figure supplement 1*). Nonetheless, hard-tuning the model could further increase the sensitivity without additional training data, such as by adjusting the temporal interval or by lowering the confidence threshold. With respect to the false positives, most were mitotic cells due to their morphological similarities with apoptotic nuclei. Nevertheless, the FPR was contained, translating into a new false positive every four frames (or 20 min of acquisition). This rate confirmed that ADeS is overall robust, especially in light of movies depicting 1700 cells per frame.

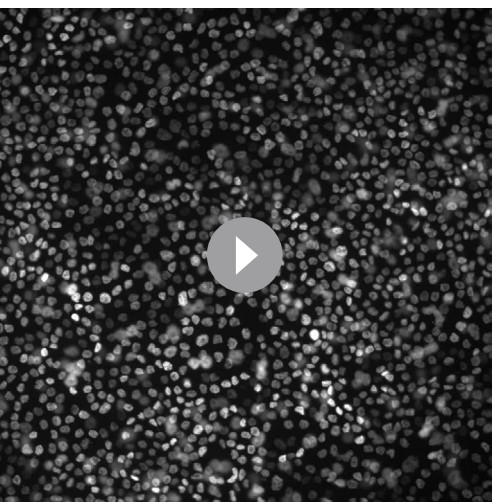

**Video 1.** Prediction of apoptotic events in vitro.
https://elifesciences.org/articles/90502/figures#video1

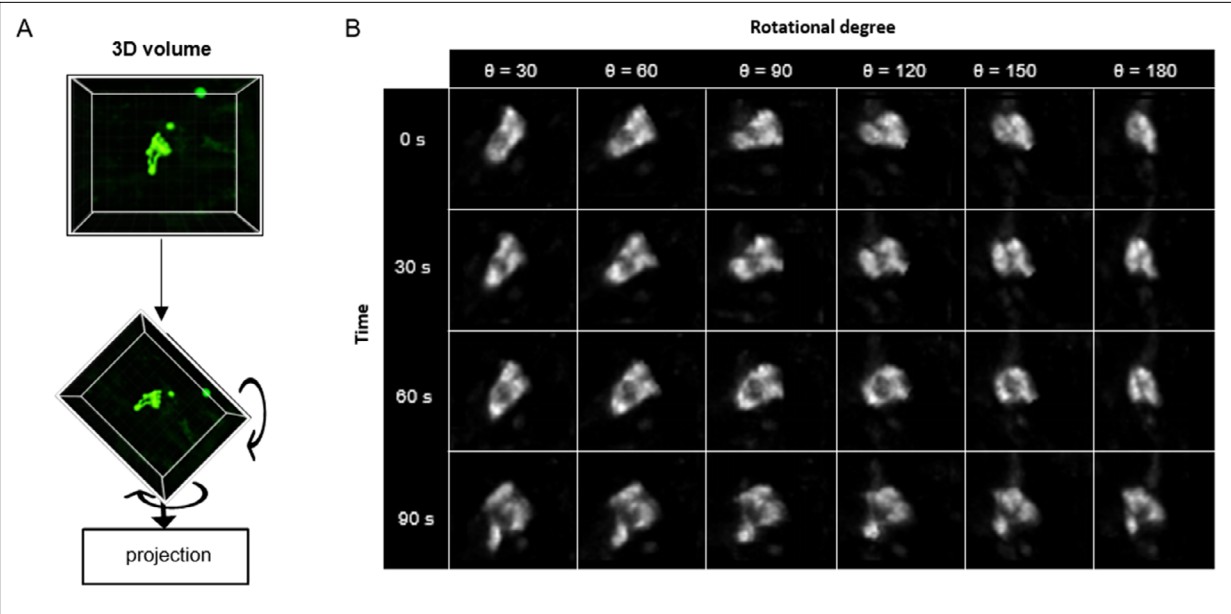

**Figure 5.** 3D rotation of the in vivo dataset. (**A**) Depiction of a 3D volume cropped around an apoptotic cell. Each collected apoptotic sequence underwent multiple 3D rotation in randomly sampled directions. The rotated 3D images were successively flattened in 2D. (**B**) Gallery showing the result of multiple volume rotations applied to the same apoptotic sequence. The vertical axis depicts the sequence over time, whereas the horizontal describes the rotational degree applied to the volumes.

Concerning the spatial–temporal dynamics, the apoptotic count over time highlighted a tight relationship between ground-truth apoptosis and correct detections of ADeS (*Figure 4H*). Accordingly, the two curves were divergent but highly correlative (Pearson $r = 0.998$), proving that ADeS can successfully capture cell death dynamics. A 2D scatterplot (x, y, t = radius; *Figure 4I*) visually depicted the spatial–temporal proximity between ADeS and the ground truth, indicating overlap between the two scatter populations. Nearest neighbor (NN) analysis further captured this relationship; the average distance between all ADeS predictions (true positives + false positives), and the NN in the ground truth was 30 pixels. In contrast, randomly generated predictions had a ground-truth NN within a 52-pixel radius (*Figure 4J*). Considering instead the true positives only, we observed that they were in close spatial proximity to the ground truth, with most predictions falling within a 20-pixel radius (*Figure 4K*). The difference between the predicted timing of apoptosis and the one annotated in the ground truth was also slight, with an average discard of 3.46 frames (*Figure 4L*). Interestingly, ADeS showed a bias toward late detections, which is explained considering that operators annotated the beginning of the apoptosis, whereas ADeS learned to detect nuclear disruption, occurring at the end of the process. Altogether, these quantifications indicate that ADeS detects apoptotic nuclei with high spatial and temporal accuracy, establishing a novel comparative baseline for this task.

### 3D rotation of the in vivo dataset

Upon the successful application of ADeS in vitro, the next step in complexity was detecting apoptosis in vivo timelapses. The latter is inherently more challenging due to different factors, including high background signal, autofluorescence, and the presence of collagen (*Pizzagalli et al., 2018*), among others. For this purpose, we retrained ADeS using the in vivo data described in *Figure 1*. However, one of the main limitations of supervised DL is the need for large datasets, and the finite number of MP-IVM acquisitions and apoptotic instances represented a bottleneck for the training of ADeS. To overcome this limitation, we implemented a custom data augmentation strategy that exploits 3D volumetric rotations, as previously performed in other studies (*Xu et al., 2020*; *Zhuang, 2019*). Accordingly, each 3D apoptotic sequence underwent multiple spatial rotations and was successively projected in 2D (*Figure 5A*). This procedure enabled us to increase the dataset of a 100-fold factor without introducing imaging artifacts as each volume rotation was a physiological representation of the cell (*Figure 5B*).

## Training and deployment in vivo

To train ADeS using the latter rotated in vivo dataset (*Figure 6*), we defined a binary classification task in which ROIs containing apoptotic cells were assigned to the class label 1. In contrast, all remaining ROIs, including healthy cells and background elements, were assigned to the class label 0 (*Figure 6—figure supplement 1A*). Subsequently, we trained the DL classifier for 200 epochs. Finally, we performed fivefold cross-validation according to the ID of the movies (*Figure 6A*). The resulting confusion matrix demonstrated a classification accuracy of 97.80% and a 2.20% misclassification rate that is primarily due to type II error (1.80% false negatives) (*Figure 6B*). Analogous to the tests in vitro, classification in vivo proved highly effective in predicting apoptotic and nonapoptotic instances. The ROC of the model, which indicated high sensitivity and a low FPR, supported this favorable result (*Figure 6C*).

We then benchmarked ADeS in the detection task performed on a set of 23 MP-IVM acquisitions of immune cells undergoing apoptosis. Unlike in vitro settings, in vivo acquisitions displayed high variability in cell number, autofluorescence, signal intensity, and noise levels (*Figure 6—figure supplement 1B*). Still, ADeS correctly predicted the location and timing of cells undergoing apoptosis (*Figure 6H*, *Video 2*), indicating its robustness to increasingly populated fields of view (*Figure 6—figure supplement 1C*). In addition, we successfully applied the pipeline to neutrophils imaged in the lymph node (*Figure 6D*) and eosinophils in the spleen (*Figure 6E*). By comparing ADeS predictions with the annotated ground truths, we found that our pipeline detected apoptotic events with a TPR of 81.3% and an FPR of 3.65% (*Figure 6F*). The detections, provided in the form of bounding boxes and trajectories, indicated the coordinates and duration of the events. Hence, to measure how close they were to the annotated trajectories, we employed the tracking accuracy metric (TRA), a compound measure that evaluates the similarities between predicted and ground-truth trajectories. The average TRA was above 0.9, indicating the high fidelity of the trajectories predicted by ADeS (*Figure 6G*).

Next, we compared ADeS to human annotation performed by three operators on five testing movies. As a result, ADeS displayed an upward trend of the TPR and a downward trend of the FPR. However, we found no significant difference in the TPR and FPR (*Figure 6H*). Regardless, ADeS performances appeared to be distributed across two distinct groups: a predominant group with an average sensitivity of 100% (>75% range) and a smaller group with an average sensitivity of 53% (41–75% range, *Figure 6H*). To understand this discrepancy, we applied hierarchical clustering to the testing videos according to their imaging properties and biological content (*Figure 6I*), thus generating two major dendrograms. The first dendrogram mostly contained videos with reduced sensitivity (yellow) and was defined by a high cell number, high noise levels, short cell distance, and a saturated and fluctuating image signal. Most notably, the cell number played a crucial role in overall performance, as reflected in the fact that an increment of this parameter resulted in a pronounced decrease in the TPR and a moderate increase in the FPR (*Figure 6J*). Incidentally, the positive predictive value (PPV) was significantly lower in videos with poor SNR and, although not statistically significant, the PPV was lower when the signal standard deviation was higher (*Figure 6K*, *Video 3*). As similar findings were observed in vitro (*Figure 4—figure supplement 2*), we hypothesized that the quality of a movie predicts ADeS performance. Hence, we combined the parameters highlighted by the clustering analysis (*Figure 6I*) into a single score ranging from 0 to 1 (1 indicating the highest and ideal score) and, in doing so, found there to be a weak correlation between the video quality and the sensitivity of ADeS (*Figure 6L*). However, this trend was evident only when we considered videos with suboptimal sensitivity; indeed, in these cases, we found a strong correlation (0.72), confirming that the video quality partially explains the observed performances (*Figure 6M*).

Finally, we evaluated how the biological variability in vivo could affect the readout of ADeS, defining nine distinct biological categories, including apoptotic cells, healthy cells, and background elements. For all biological categories, the classification accuracy was above 80%, except for overlapping cells and cells with high membrane plasticity (*Figure 6—figure supplement 1D*).

## Comparison with the state-of-the-art

To compare the performance of ADeS with other state-of-the-art algorithms for cell death quantification, we conducted a comprehensive literature review. For each study, we reported the attained classification accuracy, the experimental setup, the architecture of the classifier, the capability of detecting cell death events in movies, and the number of cell deaths in the training set (*Table 2*). Initial results

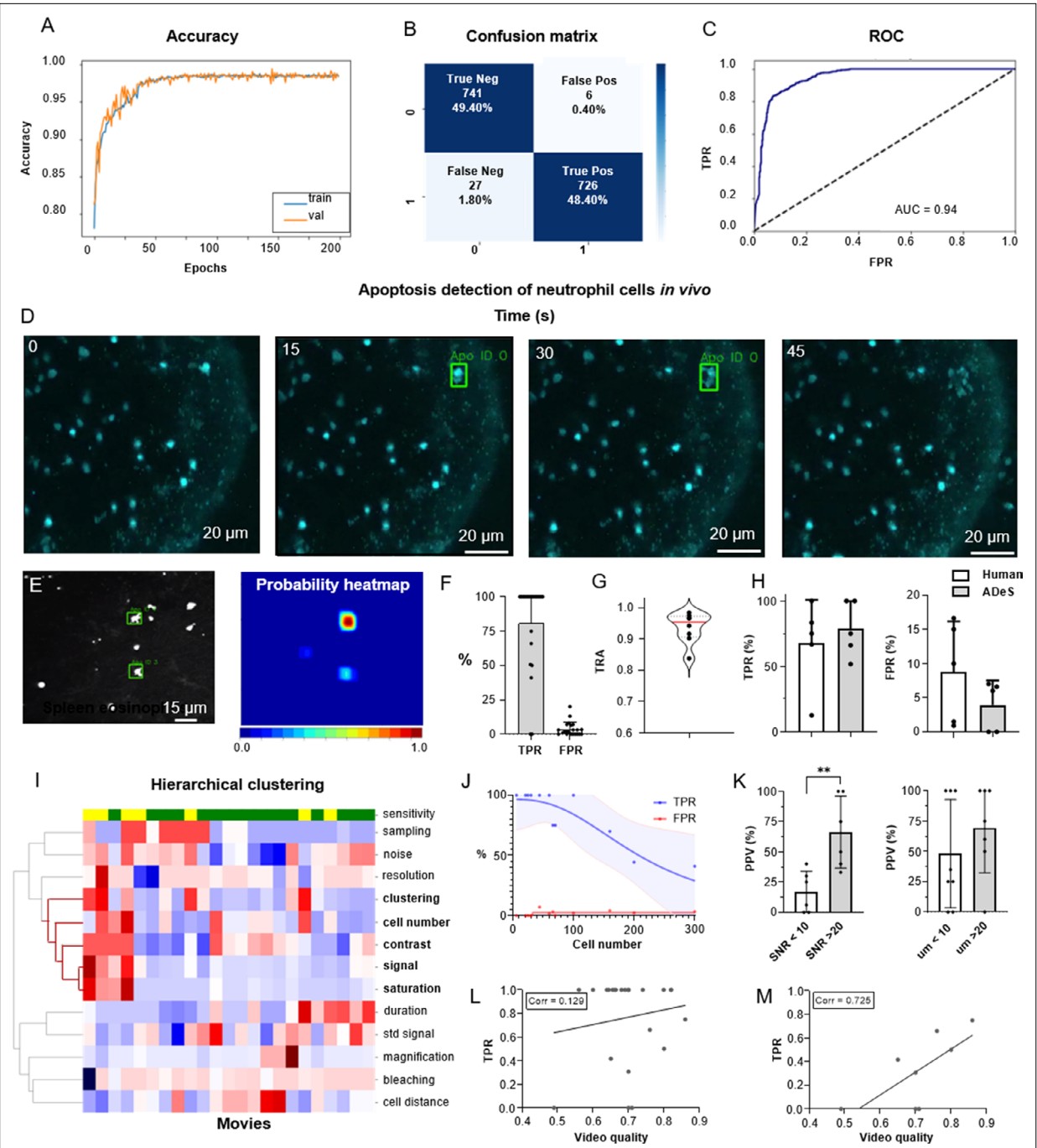

**Figure 6.** Training and performance in vivo. (**A**) Confusion matrix of the trained model at a decision-making threshold of 0.5. (**B**) Receiver-operating characteristic displaying the false positive rate (FPR) corresponding to each true positive rate (TPR). (**C**) Training accuracy of the final model trained for 200 epochs with data augmentations. (**D**) Image gallery showing ADeS classification to sequences with different disruption timing. The generated heatmap reaches peak activation (red) at the instant of cell disruption. (**D**) Representative snapshots of a neutrophil undergoing apoptosis. Green bounding boxes represents ADeS detection at the moment of cell disruption. (**E**) Representative micrograph depicting the detection of two eosinophils undergoing cell death in the spleen (left) and the respective probability heatmap (right). (**F**) ADeS performances expressed by means of TPR and FPR over a panel of 23 videos. (**G**) Tracking accuracy metric (TRA) measure distribution of the trajectories predicted by ADeS with respect to the annotated ground truth (n = 8) (**H**) Comparison between human and ADeS by means of TPR and FPR on a panel of five randomly sampled videos. (**I**) Hierarchical clustering of several video parameters producing two main dendrograms (n = 23). The first dendrogram includes videos with reduced sensitivity and is enriched in several parameters related to cell density and signal intensity. (**J**) Graph showing the effect of cell density on the performances expressed in terms of TPR and FPR (n = 13). (**K**) Comparison of the positive predictive value between videos with large and small signal-to-noise ratio (left) and videos with large and small shortest cell distance (right). (**L, M**) Selected video parameters are combined into a quality score that weakly correlates with the TPR

*Figure 6 continued on next page*

*Figure 6 continued*

in overall data (**M**, n = 23) and strongly correlates with the TPR in selected underperforming data (**N**, n = 8). Statistical comparison was performed with Mann–Whitney test. Columns and error bars represent the mean and standard deviation, respectively. Statistical significance is expressed as *p≤0.05, **p≤0.01, ***p≤0.001, ****p≤0.0001.

The online version of this article includes the following figure supplement(s) for figure 6:

**Figure supplement 1.** Training and deployment in vivo.

indicate that ADeS achieved the highest classification accuracy, but a direct comparison in terms of accuracy is not meaningful due to the differences in datasets, including distinct cell types, different types of cell death, and varying dataset sizes. For a more appropriate benchmark, we refer to *Table 1*, which shows that our classifier outperformed the baseline reimplementations of the main classifiers used in other studies.

From *Table 2*, we observe that ADeS is the only algorithm for cell death quantification that has been applied in vivo. Additionally, only ADeS and the study by *Vicar et al., 2020* effectively detected apoptotic cells in fully uncropped microscopy movies, which is a significant achievement given the computational challenge associated with the task. However, Vicary and colleagues relied on the temporal analysis of cell trajectories, while ADeS used vision-based methods to directly analyze consecutive frames of a movie. As a result, ADeS offers a comprehensive and pioneering pipeline for effectively applying vision-based classifiers to detect cell death in imaging timelapses.

## Applications for toxicity assay in vitro

A common application of cell death staining is the evaluation of the toxicity associated with different compounds (*Atale et al., 2014*; *Schmid et al., 2007*) or the efficacy of an apoptotic-inducing treatment. Here, we show that ADeS has analogous purposes and can effectively quantify the toxicity of different compounds in vitro. For this application, we grew epithelial cells in vitro, treating them with PBS and three increasing concentrations of doxorubicin, a chemotherapeutic drug that elicits apoptosis in the epithelium (*Eom et al., 2005*). Epithelial cells were seeded with the same density of cells per well, and all four conditions had the same confluence before the treatment. However, at 24 hr post-acquisition, the number of survivor cells was inversely proportional to the doxorubicin concentration (*Figure 7A*). We confirmed this trend using ADeS (*Videos 4–7*), which measured the lowest mortality after 24 hr in PBS (62 cells), followed by doxorubicin concentrations of 1.25 µM (95 cells), 2.50 µM (167 cells), and 5.00 µM (289 cells). Moreover, ADeS predicted distinct pharmacodynamics (*Figure 7B*), which can define the drug concentration and experimental duration required to reach a specific effect in the apoptotic count. To this end, each time point in *Figure 7B* also defines a dose–response relationship. Here we provide two dose–responses curves at 5 hr and 24 hr post-treatment, showing different

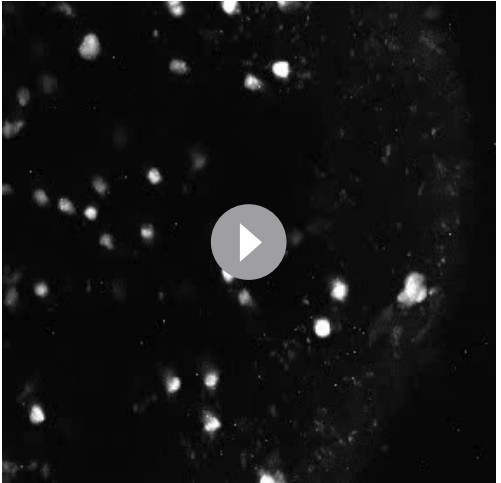

**Video 2.** Prediction of apoptotic events in vivo.
https://elifesciences.org/articles/90502/figures#video2

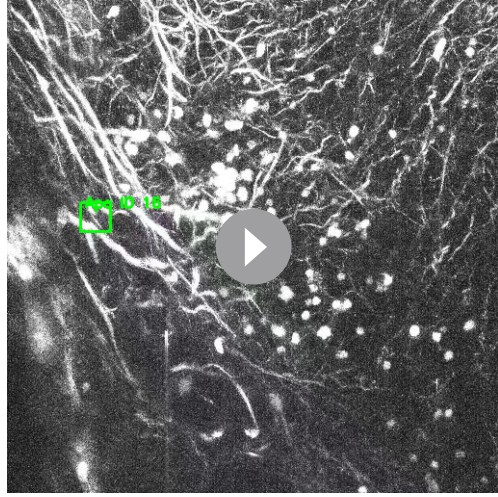

**Video 3.** Noise affects the performance of ADeS in vivo.
https://elifesciences.org/articles/90502/figures#video3

**Table 2.** Comparison of cell death identification studies.
Table reporting all studies on cell death classification based on machine learning. For each study, we included the reported classification accuracy, the experimental conditions of the studies, the target input of the classifier, and the capability of performing detection on static frames or microscopy timelapses. Met conditions are indicated with a green check. Moreover, for each study we reported the architecture of the classifier and the number of apoptotic cells in the training set. NA stands for not available and indicates that the information is not reported in the study.

| Study | Input of the classifier | Reported classification accuracy | In vitro | In vivo | Detection In frame | Detection in movies | Classifier architecture | N cell death |
|---|---|---|---|---|---|---|---|---|
| Our | Frame sequence | 98.27% | ✓ | ✓ | ✓ | ✓ | Conv-Transformer | 13,120 |
| *Jin et al., 2022* | Frame | 93% | ✓ | ✗ | ✗ | ✗ | Logistic regression | NA |
| *Verduijn et al., 2021* | Frame | 87% | ✓ | ✗ | ✗ | ✗ | VGG-19 | 19,339 |
| *Kabir et al., 2022* | Frame sequence | 93% | ✓ | ✗ | ✗ | ✗ | ResNet101-LSTM | 3172 |
| *La Greca et al., 2021* | Frame | 96.58% | ✓ | ✗ | ✗ | ✗ | ResNet50 | 11,036 |
| *Mobiny et al., 2020* | Frame sequence | 93.8% | ✓ | ✗ | ✗ | ✗ | CapsNet-LSTM | 41,000 |
| *Kranich et al., 2020* | Frame | 93.2% | ✓ | ✗ | ✗ | ✗ | CAE-RandomForest | 27,224 |
| *Vicar et al., 2020* | Frame sequence | NA | ✓ | ✗ | ✓ | ✓ | biLSTM | 1745 |
| *Jimenez-Carretero et al., 2018* | Frame | NA | ✓ | ✗ | ✓ | ✗ | R-CNN | 255,215 |

pharmacodynamics (EC50 5 hr = 2.35, Hill slope 5 hr = 3.81, EC50 24 hr = 4.47, Hill slope 24 hr = 1.93, *Figure 7C and D*). Notably, the fit can project the dose–responses for higher drug concentrations, predicting the maximum effect size at a given time. For instance, at 24 hr post treatment, a 10 µM titration attains 86% of the maximum effect (456 apoptotic cells), whereas a further increase in the concentration of the drug leads only to a moderate increase of the toxicity (*Figure 7E*). We argue that this approach helps to maximize the effect of a drug on a designated target, while minimizing collateral damage done to nontarget cells. For instance, in chemotherapies employing doxorubicin, apoptosis of epithelial cells is an undesired effect. Therefore, researchers can select a titration of the drug and a duration of the treatment that does not affect the epithelium yet still positively affects the tumor. Finally, we also demonstrated the reproducibility of the toxicity assay by targeting another cell type (T cells) treated with a different apoptotic inducer (staurosporine, *Figure 7—figure supplement 1*).

## Measurement of tissue dynamics in vivo

To test the application of ADeS in an in vivo setting, we applied it to study the response of bystander cells following apoptotic events in the lymph nodes of mice treated with an influenza vaccine. We computed the spatial and temporal coordinates of a neutrophil undergoing apoptosis (*Figure 8A*), which, combined with the tracks of neighboring cells, allowed us to characterize cellular response patterns following the apoptotic event. Among other parameters, we observed a sharp decrease in the distance between the neighboring cells and the apoptotic centroid (*Figure 8B*) in addition to a pronounced increase in the instantaneous speed of the cells (*Figure 8C*).

Successively, we evaluated the detection of apoptotic cells following laser ablation in the spleen of an anesthetized mouse (*Figure 8D*). Previous research has employed this method to study immune cell responses to tissue damage (*Uderhardt et al., 2019*). The insult caused prompt recruitment of neutrophils, leading to the formation of a local swarm (*Figure 8E*, left). After that, the neutrophils within the swarm underwent apoptotic body formation in a coordinated manner (*Figure 8E*, right). To

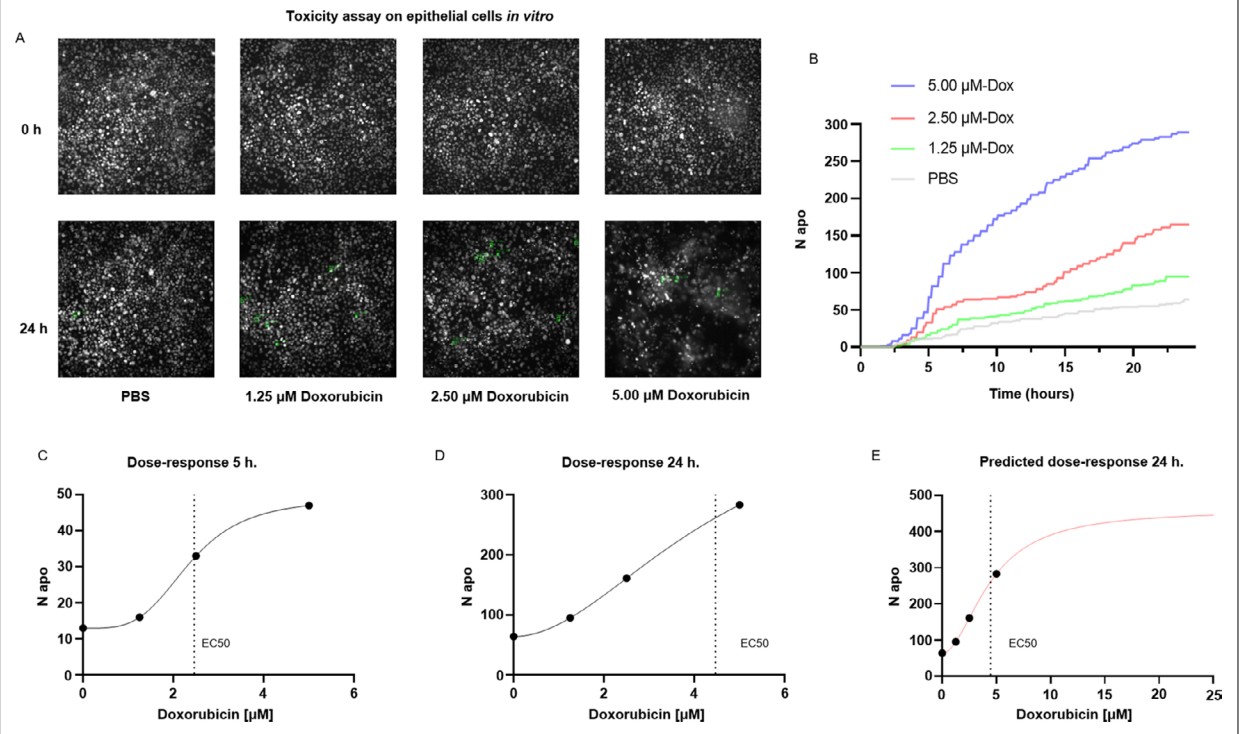

**Figure 7.** Applications for toxicity assay in vitro. (**A**) Representative snapshots depicting epithelial cells in vitro at 0 and 24 hr after the addition of PBS and three increasing doses of doxorubicin, a chemotherapeutic drug and apoptotic inducer (three replicates). (**B**) Plot showing the number of apoptotic cells detected by ADeS over time for each experimental condition. (**C, D**) Dose–response curves generated from the drug concentrations and the respective apoptotic counts at 5 hr and 24 hr post-treatment. Vertical dashed lines indicate the EC50 concentration. (**E**) Dose–response curve projected from the fit obtained in (**D**). The predicted curve allows to estimate the response at higher drug concentrations than the tested ones.

The online version of this article includes the following figure supplement(s) for figure 7:

**Figure supplement 1.** Applications for toxicity assay in vitro.

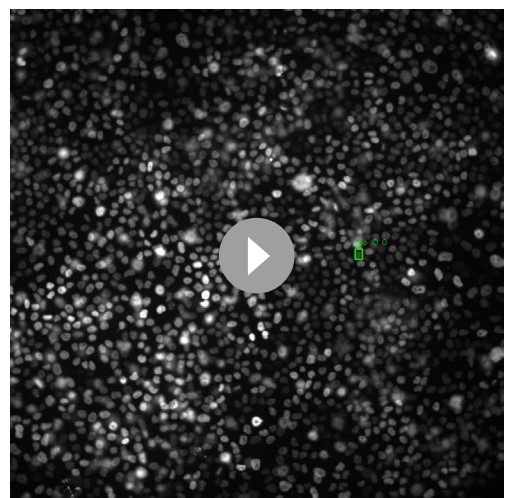

**Video 4.** In vitro detections of apoptotic cells treated with PBS for 24h.
https://elifesciences.org/articles/90502/figures#video4

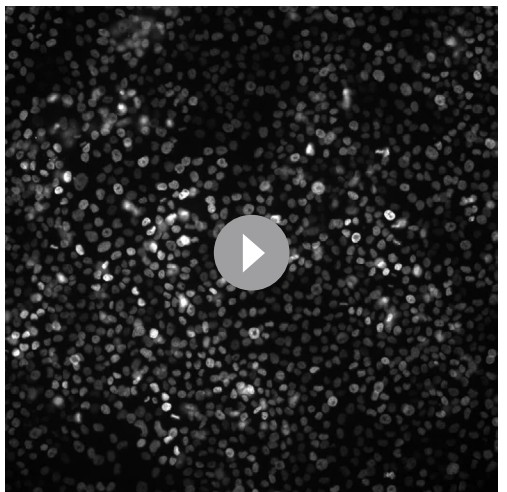

**Video 5.** In vitro detection of apoptotic cells treated with 1.25 µM doxorubicin.
https://elifesciences.org/articles/90502/figures#video5

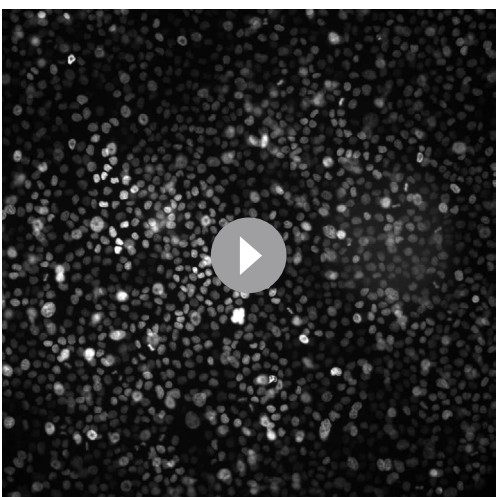

**Video 6.** In vitro detection of apoptotic cells treated with 2.50 µM doxorubicin.
https://elifesciences.org/articles/90502/figures#video6

quantify this event, we processed the generated timelapse with ADeS, resulting in a probability map of apoptotic events throughout the acquisition (x,y,t,p; *Figure 8F*). Accordingly, the location with the highest probability corresponded to the area damaged by the laser, while the visual representation of the probability map enabled us to infer the morphology and location of the swarm. This result demonstrates the potential application of ADeS in digital pathology, showing how the distribution of apoptotic events throughout the tissue can identify areas enriched by cell death events.

## Discussion

Automated bio-image analysis obviates the need for manual annotation and avoids bias introduced by the researcher. In this regard, recent studies showed the promising usage of DL to classify static images (*Jimenez-Carretero et al., 2018*; *Kranich et al., 2020*; *Verduijn et al., 2021*) or timelapses containing single apoptotic cells (*Mobiny et al., 2020*). However, these approaches are unsuitable for microscopy timelapses because they do not address two fundamental questions: the location, over the whole field of view, at which an event occurs, and its duration. These questions define a detection task (*Zhao et al., 2019*) in space and time, which has a computational cost that can rapidly grow with the size and length of a movie. Moreover, live-cell imaging data present specific challenges that further increase the difficulty of detection routines, including densely packed fields of view, autofluorescence, and imaging artifacts (*Pizzagalli et al., 2018*).

Consequently, computational tools to effectively detect apoptotic events in live-cell imaging remained unavailable. Thus, we created an apoptosis detection pipeline that could address the above-mentioned challenges in vitro and in vivo. In this regard, ADeS represents a crucial bridge between AR and bioimaging analysis, being the first apoptosis detection routine with demonstrated applicability to full microscopy timelapses. In addition, we presented two comprehensive and curated datasets encompassing multiple cell types, fluorescent labels, and imaging techniques to encourage reproducibility and foster the development of apoptosis detection routines.

In human activity recognition benchmark, 3DCNNs (*Vrskova et al., 2022*), two-streams networks (*Ye et al., 2019*), and recurrent neural networks (RNNs) (*Mohd Noor et al., 2022*) have proved to score the highest accuracy on most kinetic datasets (*Ullah et al., 2021*). However, in most studies for the classification of apoptosis, authors unanimously employed RNNs such as Conv-LSTMs. This choice, although produced valid results, is not necessarily optimal for the task. In this regard, Ullah and colleagues highlighted that the performances of different DL architectures are highly dependent on the AR dataset (*Ullah et al., 2021*). Therefore, selecting the most suitable one is only possible after an extensive benchmark. In our comparison, we demonstrated for the first time that attention-based networks are suitable for the classification and detection of apoptotic events. Accordingly,

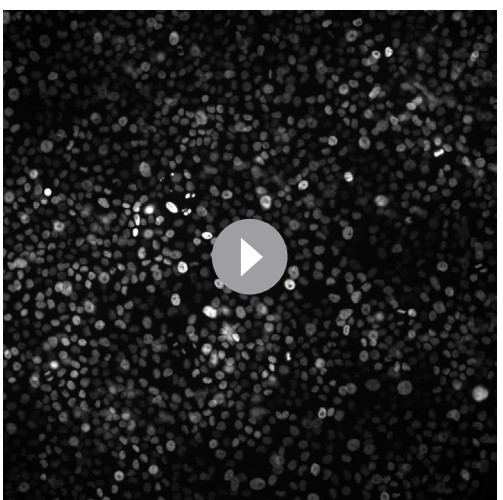

**Video 7.** In vitro detection of apoptotic cells treated with 5.00 µM doxorubicin.
https://elifesciences.org/articles/90502/figures#video7

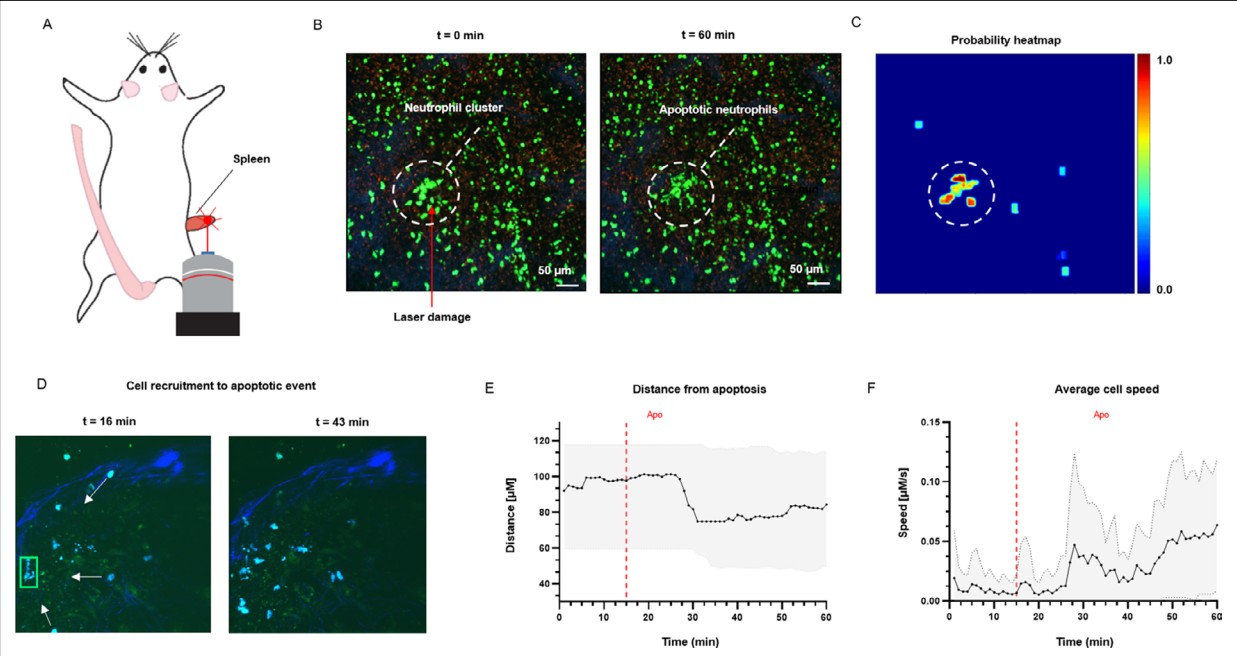

**Figure 8.** Measurement of tissue dynamics in vivo. (**A**) Intravital two-photon micrographs showing ADeS detection of an apoptotic neutrophil (blue, left) and the subsequent recruitment of neighboring cells (right) in the popliteal LN at 19 hr following influenza vaccination. (**B**) Plot showing the distance of recruited neutrophils with respect to the apoptotic coordinates over time (n = 22). (**C**) Plot showing the instantaneous speed of recruited neutrophils over time (n = 22). The dashed vertical lines indicate the instant in which the apoptotic event occurs. Gray area defines the boundaries of maximum and minimum values. (**D**) Schematic drawing showing the intravital surgical setup of a murine spleen after inducing a local laser ablation. (**E**) Intravital two-photon micrographs showing the recruitment of GFP-expressing neutrophils (green) and the formation of a neutrophil cluster (red arrows) at 60 min after photo burning induction. (**F**) Application of ADeS to the generation of a spatiotemporal heatmap indicating the probability of encountering apoptotic events in the region affected by the laser damage. The dashed circle indicates a hot spot of apoptotic events.

our Conv-Transformer network outperformed DL architectures previously employed in other studies, including 3DCNNs and RNNs. This result established a landmark in the application of attention-based networks in AR for live-cell imaging. Moreover, it suggests the possible benefits of employing transformers for the classification of different biological activities other than cell death.

Similar to most diagnostic tools, ADeS displayed a tradeoff between sensitivity (TPR) and specificity (1 – FPR), which is a known challenge in binary classification (*Pang et al., 2022*). This tradeoff can be attributed to the fact that apoptosis is rare in normal physiological conditions, leading to a high degree of class imbalance during training. As a result, the choice of the training set had a significant impact on the performances of ADeS. For instance, we highlighted the importance of a training and validation set that included challenges related to real live-cell imaging acquisitions, such as overlapping cells and low signal-to-noise samples. Including these challenges instances enabled ADeS to attain low misclassification rate and robust real-life performances. Nonetheless, we observed residual misclassifications due to shared similarities between healthy and apoptotic cells. For instance, in vitro mitotic divisions could mislead the detection of apoptotic nuclei, while in vivo, overlapping cells were sometimes mistaken for apoptotic cells. Therefore, to effectively address these challenges, it is crucial to implement strategies to increase the representativeness of the dataset, such as integrating multiple data sources and data augmentation techniques.

From a biological perspective, ADeS has multiple applications in fundamental and clinical research. Among other advantages, it can provide insights into pivotal cell death mechanisms, monitor the therapies used to modulate apoptosis in various diseases, and characterize the toxicity of different compounds. In this regard, ADeS readout is analogous to standard fluorescent probes for apoptosis detection, with the advantage that it can be applied directly to nuclear or cytoplasmic staining without the need of additional fluorescent reporters. Therefore, ADeS avoids using any additional acquisition channel, which can be used for multiplexing purposes. Moreover, common probes (*Atale et al., 2014*; *Kyrylkova et al., 2012*; *Loo, 2011*; *Sun et al., 2008*; *Vermes et al., 1995*) flag early apoptosis stages,

activated up to several minutes before the point at which morphological changes in the cell (*Green, 2005*; *Takemoto et al., 2003*); meanwhile, these cells can reverse the apoptotic process (*Geske et al., 2001*; *Masri and Chandrashekhar, 2008*; *Tang et al., 2009*). By contrast, ADeS indicates the exact instant of cell disruption, thus adding specificity to the spatial–temporal dimension. For these reasons, we suggest that ADeS can complement the information provided by classic apoptotic biomarkers, which will prove advantageous in experimental assays where the temporal resolution delivers more information than the sole apoptotic count. Moreover, ADeS can be usefully applied in processing high-throughput live-cell imaging, minimizing annotation time and research bias.

Finally, in tissue dynamics the spatial–temporal activity of cells can reveal connections between signaling pathways and the fate decision of individual cells, such as mitosis or apoptosis (*Gagliardi et al., 2021*). These intricate systems can display complex dynamics, which can be better comprehended incorporating spatial and temporal coordinates provided by ADeS. Consequently, we propose that integrating these spatial–temporal characteristics with experimental observations could lay the groundwork for understanding the mechanism governing complex signaling pathways. Furthermore, we contend that this information has the potential to facilitate the development of predictive models, establishing a correlation between specific cell death dynamics and the underlying stimuli. This, in turn, could serve as the foundation for innovative diagnostic tools capable of inferring the cause of cell death (*Fesik, 2005*; *Hotchkiss and Nicholson, 2006*).

In conclusion, ADeS constitutes a novel solution for apoptosis detection that combines state-of-the-art microscopy and DL. Its successful implementation represents a step toward the general application of AR methods to live-cell imaging. By bridging these two distinct fields, ADeS leverages successfully the benefits of automated routines. Further work could expand the proposed pipeline to encompass diverse cell populations, various types of cell death, and potentially broader cellular activities.

# Materials and methods
## MCF10A cell line and image acquisition
The normal-like mammary epithelial MCF10A cells (provided by Joan Brugge; *Debnath et al., 2003*), stably expressing the nuclear marker, were generated as previously described (*Gagliardi et al., 2021*). Briefly, the nuclear marker H2B-miRFP703, provided by Vladislav Verkhusha (Addgene plasmid #80001) (*Shcherbakova et al., 2016*), was subcloned in the PiggyBac plasmid pPBbSr2-MCS. After cotransfection with the transposase plasmid (*Yusa et al., 2011*), cells were selected with 5 μg/ml Blasticidin and subcloned. For time-lapse imaging, the cells were seeded on 5 μg/ml fibronectin (PanReac AppliChem)-coated 1.5 glass-bottom 24-well plates (Cellvis) at $1 \times 10^5$ cells/well density. After 48 hr, when the optical density was reached, the confluent cell monolayer was acquired every 1 or 5 min for several hours with a Nikon Eclipse Ti inverted epifluorescence microscope with 640 nm LED light source, ET705/72m emission filter, and a Plan Apo air 203 (NA 0.8) or a Plan Apo air 403 (NA 0.9) objectives. The collection of biological experiments used in this study includes different stimulation of apoptosis, such as growth factors, serum starvation, and doxorubicin at various concentrations.

## Apoptosis induction of MCF10A cells with doxorubicin
Normal-like mammary epithelial MCF10A cells were grown in 24-well glass coated with fibronectin with a seeding of $1 \times 10^5$ cells/well. After 2 d, cells were starved for 3 hr and treated with doxorubicin at 1.25, 2.50, and 5.00 μM concentrations.

## Mice
Prior to imaging, LysM-cre-GFP mice were anesthetized with a cocktail of ketamine (100 mg/kg) and xylazine (10 mg/kg) as previously described (*Sumen et al., 2004*). All animals (females between 6 and 12 mo) were maintained in specific pathogen-free facilities at the Institute for Research in Biomedicine (Bellinzona, CH). All experimental procedures were performed according to the regulations of the local authorities and approved by the Swiss Federal Veterinary Office (licensing national number 39015).

## Intravital two-photon microscopy
Surgery in the popliteal lymph node was performed as previously reported (*Miller et al., 2004*). The exposed organs were imaged on a custom up-right two-photon microscope (TrimScope, LaVision

BioTec). Probe excitation and tissue second-harmonic generation were achieved with two Ti:sapphire lasers (Chamaleon Ultra I, Chamaleon Ultra II, Coherent) and an optical oscillator that emits in the 1010–1340 nm range (Chamaleon Compact OPO, Coherent) and has an output wavelength between 690 and 1080 nm.

## Neutrophil isolation from mouse bone marrow

Bone marrow samples were extracted via flushing with PBS from the long bones of UBC-GFP mice (https://www.jax.org/strain/004353). Then, the bone marrow was filtered through a 40 um strainer and resuspended in PBS. Primary bone marrow neutrophils were isolated with Ficoll gradient and resuspended in PBS.

## T-cell culture in a 3D collagen matrix

Human CD4+ T cells were isolated from the PBMC fraction of healthy donors obtained from NetCAD (Canadian Blood Services). Cell purity was above 95%. Naïve CD4+ T cells were activated by adding Dynabeads coated with anti-human CD3e/CD28 antibody (1:1 bead:cell ratio, Life Technologies, Cat# 11131D) in RPMI1640 supplemented with 10% FBS (VWR Seradigm, Cat# 1500-500), 2 mM GlutaMAX (Gibco, Cat# 3050-061), 1 mM sodium pyruvate (Corning, Cat# 25-000CI), and 10 mM HEPES (Sigma-Aldrich, Cat# H4034). After 2 d, beads were removed and cells were cultured for another 4–6 d in a medium containing 50 IU/ml human rIL-2 (Biotechne, Cat# 202-IL-500), keeping cell density at $2 \times 10^5$ cells/ml. Cells were used for all experiments between days 6–8. All work with human blood has been approved by the University of Manitoba Biomedical Research Ethics Board (BREB).

## Apoptosis live-cell imaging of T cells in 3D collagen chambers

T cells were labeled at days 6–8 using CMAC (10 µM) cell tracker dye (Invitrogen), and glass slide chambers were constructed as previously described (*Lopez et al., 2019*; *Lopez et al., 2022*). Briefly, $2 \times 10^6$ cells were mixed in 270 µl of bovine collagen (Advanced Biomatrix, Cat# 5005-100ML) at a final concentration of 1.7 mg/ml. Collagen chambers were solidified for 45 min at 37°C/5% $CO_2$ and placed onto a custom-made heating platform attached to a temperature control apparatus (Werner Instruments). For the induction of apoptosis, 1 µM of staurosporine (Sigma, Cat# 569397-100UG) and 800 ng of TNF-a (BioLegend, Cat# 570104) in 100 µl RPMI were added on top of the solidified collagen. Cells were imaged as soon as the addition of apoptosis inducers using a multiphoton microscope with a Ti:sapphire laser (Coherent), tuned to 800 nm for optimized excitation of CMAC. Stacks of 13 optical sections (512 × 512 pixels) with 4 mm z-spacing were acquired every 15 s to provide imaging volumes of 44 mm in depth (with a total time of 60–120 min). Emitted light was detected through 460/50 nm, 525/70 nm, and 595/50 nm dichroic filters with non-descanned detectors. All images were acquired using the 20 × 1.0 N.A. Olympus objective lens (XLUMPLFLN; 2.0 mm WD).

## Data processing and image analysis

The raw video data, composed by uint8 or uint16 TIFFs, were stored as HDF5 files. No video preprocessing was applied to the raw data before image analysis. Cell detection, tracking, and volumetric reconstruction of microscopy videos were performed using Imaris (Oxford Instruments, v9.7.2). The resulting data were further analyzed with custom MATLAB and Python scripts (see 'Code availability' section).

## Apoptosis annotation of epithelial MCf10A cells in vitro

We manually annotated apoptotic events of MCF10A cells by visual inspection of the movies. The annotation was done by observing the morphological changes associated with apoptosis (e.g., nuclear shrinkage, chromatin condensation, epithelial extrusion, nuclear fragmentation) across multiple consecutive frames. Using a custom Fiji (*Schindelin et al., 2012*) macro, we automatically stored x and y centroids of the apoptotic nucleus. The time *t* of each apoptotic annotation was defined as the beginning of nuclear shrinkage.

## Generation of thein vitro training dataset

The 16-bit raw movies were min-max scaled to the 0.001 and 0.999 quantiles and downsampled to 8-bit resolution. Using the database of manually labeled coordinates of apoptotic events (x,y,t),

we extracted crops with 59 × 59 pixels resolution (2× scaling for the FOV acquired with the 20× objective). Seven time steps of the same location were extracted, with linear spacing from –10 min to +50 min relative to the apoptosis annotation. This time frame was chosen to capture the cell before the onset of apoptosis, and the morphological changes associated with apoptosis (nuclear shrinkage, decay into apoptotic bodies, extrusion from epithelium). The resulting image cube has dimensions of 59 × 59 × 7. To create the training data for the nonapoptotic class, we excluded areas with an annotated apoptotic event with a safety margin from the movies. From the remaining regions without apoptoses, we extracted image cubes from cells detected with StarDist (*Fazeli et al., 2020*) and from random locations. The random crops also included debris, apoptotic bodies from earlier apoptotic events, empty regions, and out-of-focus nuclei.

## Apoptosis annotation of leukocyte cells in vivo

Three operators independently annotated the videos based on selected morphological criteria. To label apoptotic cells, the annotators considered only the sequences of cells that displayed membrane blebbing followed by apoptotic bodies formation and cell disruption (*Figure 2B*). For each frame in the apoptotic sequence, the operators placed a centroid at the center of the cell with the Imaris 'Spots' function, generating an apoptotic track. Successively, ground-truth tracks were generated according to a majority voting system, and 3D volume reconstruction was performed on ground-truth cells using the Imaris 'Surface' function. Nearby nonapoptotic cells were also tracked. In addition, other nonapoptotic events were automatically subsampled from regions without apoptotic cells.

## 3D rotation of the in vivo annotations

In vivo annotations presented a class unbalance in favor of nonapoptotic cells, with a relative few apoptotic instances. Hence, to compensate for this bias, we produced several representations of the raw data by interpolating the raw image stacks in 3D volumes and rotating them in randomly sampled directions, with rotational degrees between 0° and 45°. After each manipulation, the rotated volume underwent flattening by maximum projection and symmetric padding to preserve the original dimension. The 2D images were successively resized and cropped to match the 59 × 59 pixels input of the classifier. Finally, the training sequences were saved as uint8 grayscale TIFF files.

## Generation of the in vitro and in vivo training datasets

To detect apoptotic cells in microscopy acquisitions, we defined a 2D binary classification task in which apoptotic events are labeled with class 1, while nonapoptotic events belonged to the class label 0. The resulting unprocessed data consisted of frame sequences composed of 3D crops. The content of the class label 0 in vitro included healthy nuclei, background, cell debris, and mitotic cells. The content of the class label 0 in vivo included motile cells, arrested cells, highly deformed cells, overlapping cells, cell debris or blebs, empty background, noisy background, and collagen.

## Data augmentation and data loader

Given the varying length of the training sequences contained in the TIFFs, upon training, we used a custom data loader that uniformly samples the input data and produces sequences with a fixed number of frames. The fixed number of frames was set to 5, corresponding to the frame length of the shortest apoptotic sequence. During training, each sample underwent horizontal shift, vertical shift, zoom magnification, rotation, and flipping. All data augmentations were performed in Python using the Keras library.

## Deep learning architecture

As a deep learning classifier, we employed a custom architecture relying on time-distributed convolutional layers stacked on top of a transformer module (Conv-Transformer). The input size consists of five single-channel images with 59 × 59 pixel size. The convolutional network has three layers of size 64, 128, and 256 length. Each layer has a 3 × 3 kernel, followed by Relu activation, batch normalization, and a dropout set to 0.3. The inclusion of padding preserves the dimension of the input, while 2D max pooling is at the end of each convolutional block. After 2D max pooling, the output is passed to a transformer module counting six attention heads, and successively to a fully connected decision layer. The fully connected network has four layers with 1024, 512,128, and 64 nodes, each one followed by

Relu activation and a 0.3 dropout layer. The last layer is a softmax activation, which predicts a decision between the two classes.

## Training and hyperparameters

Our model was trained in TensorFlow with Adam optimizer using binary cross-entropy loss and an initial learning rate of 0.0001. The optimal mini-batch size was 32, and the number of training epochs was 200. In training mode, we set a checkpoint to save the model with the best accuracy on the validation dataset, and a checkpoint for early stopping with patience set to 15 epochs. In addition, the learning rate decreased when attending a plateau.

## ADeS deployment

For the deployment of the classifier on microscopy videos, we generative region proposals using the selective search algorithm, obtaining a set of ROIs for each candidate frame of the input movie. For each ROI computed by the region proposal at time $t$, a temporal sequence is cropped around $t$ and classified with the Conv-Transformer. The resulting bounding boxes are filtered according to a probability threshold and processed with the non-maxima suppression utils from Pytorch. Consecutive bounding boxes classified as apoptotic are connected using a custom multiobject tracking algorithm based on Euclidean distance. The generated trajectories are filtered by discarding those with less than two objects.

## Default and user-defined parameters

ROIs detected with the region proposal are filtered according to their size, discarding the ones with edges below 20 pixels and above 40 pixels. Furthermore, a threshold on intensity is applied to exclude uint8 patches with an average brightness below 40. Upon classification, a temporal window corresponding to the expected duration of the apoptotic event is set by the user (nine frames by default). This temporal window is subsampled to match the number of input frame of the classifier (five). The filtering of the predictions depends on a user-specified threshold, which by default corresponds to 0.95 in vivo and 0.995 in vitro. Non-maxima suppression is based on the overlapping area between bounding boxes, set to 0.1 by default. The centroid tracking has the following adjustable parameters: gap and distance threshold. The 'gap' parameter, set to three frames, specifies for how long a centroid can disappear without being attributed a new ID upon reappearance. A threshold on the distance, set by default to 10 pixels, allows the connection of centroids within the specified radius. All the reported quantifications had default parameters.

## Statistical analyses

Statistical comparisons and plotting were performed using GraphPad Prism 8 (GraphPad, La Jolla, USA). All statistical tests were performed using nonparametric Kruskal–Wallis test or Mann–Whitney test For significance, p value is represented as *p<0.05, **p<0.005, and ***p<0.0005.

## Acknowledgements

We thank Dr. Coral Garcia (IQS, Barcelona, Spain) for the support in generating graphical content. Moreover, we would like to acknowledge Gabriele Abbate (IDSIA, Lugano, Switzerland) for his help during an early implementation of the DL classifier. Suisse National Science Foundation grant 176124 (AP, DU, MP, SG); Swiss Cancer League grant KLS-4867-08-2019, Suisse National Science Foundation grant Div3; 310030_185376 and IZKSZ3_62195, Uniscientia Foundation (PG, LH, OP); SystemsX.ch grant iPhD2013124 (DU, RK, SG); Novartis Foundation for medical-biological Research, The Helmut Horten Foundation, SwissCancer League grant KFS-4223-08-2017-R (PA, MT); Canadian Institute for Health Research (CIHR) Project grants PJT-155951 (RZ, PL, TM); NCCR Robotics program of the Swiss National Science Foundation (AG, LG); Biolink grant 189699 (DU, PC, SG)

## Additional information

### Funding

| Funder | Grant reference number | Author |
|---|---|---|
| Schweizerischer Nationalfonds zur Förderung der Wissenschaftlichen Forschung | 176124 | Santiago Fernandez Gonzalez |
| Swiss Cancer League | KLS-4867-08-2019 | Olivier Pertz |
| System X | iPhD2013124 | Santiago Fernandez Gonzalez |
| Swiss Cancer League | KFS-4223-08-2017-R | Marcus Thelen |
| CIHR Skin Research Training Centre | Project grants PJT-155951 | Thomas T Murooka |
| Suisse National Science Foundation | 310030_185376 | Paolo Armando Gagliardi Lucien Hinderling Olivier Pertz |
| Suisse National Science Foundation | IZKSZ3_62195 | Paolo Armando Gagliardi Lucien Hinderling Olivier Pertz |
| Uniscientia Foundation | | Paolo Armando Gagliardi Lucien Hinderling Olivier Pertz |
| Novartis Foundation for Medical-Biological Research | | Paola Antonello Marcus Thelen |
| The Helmut Horten Foundation | | Paola Antonello Marcus Thelen |
| NCCR Robotics program of the Swiss National Science Foundation | | Alessandro Giusti Luca Maria Gambardella |
| Biolink | 189699 | Diego Ulisse Pizzagalli Pau Carrillo-Barberà Santiago Fernandez Gonzalez |

The funders had no role in study design, data collection and interpretation, or the decision to submit the work for publication.

### Author contributions

Alain Pulfer, Conceptualization, Resources, Data curation, Software, Formal analysis, Supervision, Validation, Visualization, Methodology, Writing – original draft, Writing – review and editing; Diego Ulisse Pizzagalli, Conceptualization, Supervision, Writing – review and editing; Paolo Armando Gagliardi, Data curation, Validation, Investigation, Visualization, Writing – review and editing; Lucien Hinderling, Data curation, Formal analysis, Visualization, Methodology; Paul Lopez, Miguel Palomino-Segura, Data curation, Methodology; Romaniya Zayats, Visualization, Methodology; Pau Carrillo-Barberà, Data curation, Methodology, Writing – original draft; Paola Antonello, Methodology; Benjamin Grädel, Software, Visualization; Mariaclaudia Nicolai, Software; Alessandro Giusti, Supervision, Visualization; Marcus Thelen, Luca Maria Gambardella, Rolf Krause, Supervision; Thomas T Murooka, Olivier Pertz, Supervision, Writing – review and editing; Santiago Fernandez Gonzalez, Conceptualization, Supervision, Funding acquisition, Methodology, Writing – original draft, Project administration, Writing – review and editing

### Author ORCIDs

Alain Pulfer ⓘ https://orcid.org/0009-0004-3780-1642
Paolo Armando Gagliardi ⓘ https://orcid.org/0000-0002-4818-035X

Lucien Hinderling https://orcid.org/0000-0002-3956-9363
Miguel Palomino-Segura https://orcid.org/0000-0003-1614-1222
Benjamin Grädel https://orcid.org/0000-0002-1995-0263
Marcus Thelen https://orcid.org/0000-0002-3443-1605
Olivier Pertz https://orcid.org/0000-0001-8579-4919
Santiago Fernandez Gonzalez https://orcid.org/0000-0003-4166-7664

Reviewer #1 (Public Review): https://doi.org/10.7554/eLife.90502.3.sa1
Reviewer #2 (Public Review): https://doi.org/10.7554/eLife.90502.3.sa2
Author response https://doi.org/10.7554/eLife.90502.3.sa3

## Additional files

### Supplementary files
MDAR checklist

### Data availability
All data are uploaded in the following repositories: Imaging data: https://doi.org/10.5281/zenodo.10820761 Code: https://github.com/mariaclaudianicolai/ADeS (copy archived at *Nicolai, 2024*).

The following dataset was generated:

| Author(s) | Year | Dataset title | Dataset URL | Database and Identifier |
|---|---|---|---|---|
| Pulfer A | 2024 | Transformer-based spatial-temporal detection of apoptotic cell death in live-cell imaging | https://doi.org/10.5281/zenodo.10820761 | Zenodo, 10.5281/zenodo.10820761 |

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
