## [Editor Report · eLife assessment]

This **valuable** study advances our understanding of spatial–temporal cell dynamics both in vivo and in vitro. The authors provide **solid** evidence for their innovative deep learning-based apoptosis detection system, ADeS, which utilizes the principle of activity recognition. This work will be of broad interest to cell biologists and neuroscientists.

---

## [Referee Report · Reviewer #1 (Public Review)]

Summary:

Pulfer et al., describes the development and testing of a transformer based deep learning architecture called ADeS, which the authors use to identify apoptotic events in cultured cells and live animals. The classifier is trained on large datasets and provides robust classification accuracies in test sets that are comparable to and even outperform existing deep learning architectures for apoptosis detection. Following this validation, the authors also design use cases for their technique both in vitro and in vivo, demonstrating the value of ADeS to the apoptosis research space.

Strengths:

ADeS is a powerful tool in the arsenal of cell biologists interested in the spatio-temporal co-ordinates of apoptotic events in vitro, since live cell imaging typically generates densely packed fields of view that are challenging to parse by manual inspection. The authors also integrate ADeS into the analysis of data generated using different types of fluorescent markers in a variety of cell types and imaging modalities, which increases its adaptability by a larger number of researchers. ADeS is an example of successful deployment of activity recognition (AR) in the automated bioimage analysis space, highlighting the potential benefits of AR to quantifying other intra- and intercellular processes observable using live cell imaging.

Weaknesses:

A major drawback was the lack of access to the ADeS platform for the reviewers; the authors state that the code is available in the code availability section, which is missing from the current version of the manuscript. This prevented an evaluation of the usability of ADeS as a resource for other researchers. The authors also emphasize the need for label-free apoptotic cell detection in both their abstract and their introduction but have not demonstrated the performance of ADeS in a true label-free environment where the cells do not express any fluorescent markers. While Pulfer et al., provides a wealth of information about the generation and validation of their DL classifier for in vitro movies, and the utility of ADeS is obvious in identifying apoptotic events among FOVs containing ~1700 cells, the evidence is not as strong for in vivo use cases. They mention the technical challenges involved in identifying apoptotic events in vivo, and use 3D rotation to generate a larger dataset from their original acquisitions. However, it is not clear how this strategy would provide a suitable training dataset for understanding the duration of apoptotic events in vivo since the temporal information remains the same. The authors also provide examples of in vivo acquisitions in their paper, where the cell density appears to be quite low, questioning the need for automated apoptotic detection in those situations. In the use cases for in vivo apoptotic detection using ADeS (Fig 8), it appears that the location of the apoptotic event itself was obvious and did not need ADeS, as in the case of laser ablation in the spleen and the sparse distribution of GFP labeled neutrophils in the lymph nodes. Finally, the authors also mention that video quality altered the sensitivity of ADeS in vivo (Fig 6L) but fail to provide an example of ADeS implementation on a video of poor quality, which would be useful for end users to assess whether to adopt ADeS for their own live cell movies.

---

## [Referee Report · Reviewer #2 (Public Review)]

Summary:

Pulfer A. et al. developed a deep learning-based apoptosis detection system named ADeS, which outperforms the currently available computational tools for in vitro automatic detection. Furthermore, ADeS can automatically identify apoptotic cells in vivo in intravital microscopy time-lapses, preventing manual labeling with potential biases. The authors trained and successfully evaluated ADeS in packed epithelial monolayers and T cells distributed in 3D collagen hydrogels. Moreover, in vivo, training and evaluation were performed on polymorphonucleated leukocytes in lymph nodes and spleen.

Strengths:

Pulfer A. et colleagues convincingly presented their results, thoroughly evaluated ADeS for potential toxicity assay, and compared its performance with available state-of-the-art tools.

Weaknesses:

The use of ADeS is still restricted to samples where cells are fluorescently labeled either in the cytoplasm or in the nucleus, which limits its use for in vitro toxicity assays that are performed on primary cells or organoids (e.g., iPSCs-derived systems) that are normally harder to transfect.

In conclusion, ADeS will be a useful tool to improve output quality and accelerate the evaluation of assays in several research areas with basic and applied aims.

---

## [Author Response]

The following is the authors’ response to the original reviews.

Major changes:

Removed any claim of label-free detection, clarifying that ADeS can predict apoptotic events without apoptotic probes

Provided a github repository with the executable code (https://github.com/mariaclaudianicolai/ADeS)

Uploaded all imaging data used to train and benchmark ADeS on Zenodo ( https://doi.org/10.5281/zenodo.10820761)

Added supplementary movie showing degraded performance on noisy movie in vivo (Supplementary Movie 3)

Generated a supplementary figure showing the effect of noise on prediction accuracy (Supplementary Figure 4)

Minor changes:

Line 6: added Benjamin Grädel and Mariaclaudia Nicolai to the list of authors

Line 44: dynamics

Line 54: updated reference to a published paper

Line 65: fixed spelling of "chronic"

Line 74: fixed spelling of "limitations"

Line 76: changed “biochemical reporters” to “fluorescent probes”

Line 77: changed “label-free” to “probe-free”

Line 85: “can apply” to "can be applied"

Line 109: The citation is updated to appear in the reference

Lines 143-144: Fixed statement about apoptotic cells having non-significant displacement compared to arrested cells

Line 156: Figure 3 is cited

Line 185 and Fig 3 legends: “chore” to "core"

Lines 187 and 248: “withouth” to "without"

Lines 177-178: introduced acronyms for deep learning networks

Lines 276-277: Added interval ranges to clarify subgroups observed in Figure 6F

Line 284: substituted “SNR” with “signal-to-noise ratio”

Line 286: mentioned “Supplementary Movie 3”

Line 515: explicitly defined “field of view” instead of “FOVs”

Lines 604-606: Added data availability section

Line 822: modified caption of Figure 1D to explain the estimation of nuclear area over time

Lines 911-912: Explained gray area in caption of figure 8B-C

Supplementary figure 1: removed “Neu” and “Eos” acronyms from caption. Introduced definition of “FOV” and “SNR” acronyms

**Editorial assessment**
This valuable work by Pulfer et al. advances our understanding of spatial-temporal cell dynamics both in vivo and in vitro. The authors provide convincing evidence for their innovative deep learning-based apoptosis detection system, ADeS, that utilizes the principle of activity recognition. Nevertheless, the work is incomplete due to the authors' claim that their system is valid for non-fluorescently labeled cells, without evidence supporting this notion. After revisions, this work will be of broad interest to cell biologists and neuroscientists

We acknowledge that the “label-free” claim was misleading, and in the revised manuscript we addressed this aspect by stating that ADeS is “probe-free”, not requiring any apoptotic marker. For this reason we kindly ask the editor to modify its assessment concerning the work being incomplete, as our tool was specifically meant for fluorescent microscopy.

**Reviewer #1 (Public Review):**
Summary:Pulfer et al., describe the development and testing of a transformer-based deep learning architecture called ADeS, which the authors use to identify apoptotic events in cultured cells and live animals. The classifier is trained on large datasets and provides robust classification accuracies in test sets that are comparable to and even outperform existing deep learning architectures for apoptosis detection. Following this validation, the authors also design use cases for their technique both in vitro and in vivo, demonstrating the value of ADeS to the apoptosis research space.Strengths:ADeS is a powerful tool in the arsenal of cell biologists interested in the spatio-temporal co-ordinates of apoptotic events in vitro, since live cell imaging typically generates densely packed fields of view that are challenging to parse by manual inspection. The authors also integrate ADeS into the analysis of data generated using different types of fluorescent markers in a variety of cell types and imaging modalities, which increases its adaptability by a larger number of researchers. ADeS is an example of the successful deployment of activity recognition (AR) in the automated bioimage analysis space, highlighting the potential benefits of AR to quantifying other intra- and intercellular processes observable using live cell imaging.Weaknesses:A major drawback was the lack of access to the ADeS platform for the reviewers; the authors state that the code is available in the code availability section, which is missing from the current version of the manuscript. This prevented an evaluation of the usability of ADeS as a resource for other researchers.

We acknowledge that having access to the code is pivotal, and therefore in this revised version we deposited the Python code deploying our DL model on github (link). Moreover, we included in the revised manuscript the training datasets (in vitro and in vivo), as well as all the testing videos used to benchmark ADeS.

The authors also emphasize the need for label-free apoptotic cell detection in both their abstract and their introduction but have not demonstrated the performance of ADeS in a true label-free environment where the cells do not express any fluorescent markers.

The system was developed to primarily analyze data acquired via fluorescent microscopy, which relies on fluorescent staining to visualize cells. Therefore, it is not possible to evaluate our methodology in a 100% label-free environment. What we meant using the term “label-free” is that our method can detect apoptotic events based exclusively on morphological cues, without the use of fluorescent apoptotic reporters. We acknowledge that this terminology was misleading and we apologize for the misunderstanding. To amend this, in our revised paper we avoid using the term “label-free”, referring instead to “probe-free” detection.

While Pulfer et al., provide a wealth of information about the generation and validation of their DL classifier for in vitro movies, and the utility of ADeS is obvious in identifying apoptotic events among FOVs containing ~1700 cells, the evidence is not as strong for in vivo use cases. They mention the technical challenges involved in identifying apoptotic events in vivo, and use 3D rotation to generate a larger dataset from their original acquisitions. However, it is not clear how this strategy would provide a suitable training dataset for understanding the duration of apoptotic events in vivo since the temporal information remains the same.

One of the main challenges encountered in vivo was the difficulty of capturing rare events such as apoptosis in physiological conditions. Moreover the lack of publicly available datasets further prevented us from collecting an extended training dataset suitable for data-hungry techniques such as supervised deep learning. Resorting to 3D rotations was a strategy to exploit the visual information within acquisition volumes to train our classifiers for 2D detection. This approach is a common data augmentation technique that can naturally increment the size of a dataset by displaying the same object from different angles. However this technique does not explicitly address temporal aspects of the apoptotic events, such as their duration. The duration of the apoptotic events was empirically estimated to obtain a temporal window suitable for detection (Supplementary Figure 1K-L).

The authors also provide examples of in vivo acquisitions in their paper, where the cell density appears to be quite low, questioning the need for automated apoptotic detection in those situations. In the use cases for in vivo apoptotic detection using ADeS (Fig 8), it appears that the location of the apoptotic event itself was obvious and did not need ADeS, as in the case of laser ablation in the spleen and the sparse distribution of GFP labeled neutrophils in the lymph nodes.

Before addressing the need for these methodologies in vivo, we provide a proof of concept for their applicability. Accordingly, in vivo acquisitions present several visual artifacts and challenges that can hamper activity recognition techniques. Therefore, from a computer vision perspective, the successful implementation of ADeS in vivo is an achievement per se.

Concerning its need, we showed in supplementary figure 3 that ADeS is robust to increasingly populated fields of view, and might be useful in detecting hindered apoptotic events as well as in reducing human-bias.

Finally, the authors also mention that video quality altered the sensitivity of ADeS in vivo (Fig 6L) but fail to provide an example of ADeS implementation on a video of poor quality, which would be useful for end users to assess whether to adopt ADeS for their own live cell movies.

In figure 6L we quantitatively showed that videos affected by low quality were negatively affecting the sensitivity of ADeS. In this revised version we included a supplementary movie (supplementary movie X) depicting ADeS performances in high signal-to-noise conditions. We also addressed this aspect in vitro, by generating a synthetic degradation of the movie quality and measuring the effect on the performances (supplementary figure 4).

**Reviewer #2 (Public Review):**
Summary:Pulfer A. et al. developed a deep learning-based apoptosis detection system named ADeS, which outperforms the currently available computational tools for in vitro automatic detection. Furthermore, ADeS can automatically identify apoptotic cells in vivo in intravital microscopy time-lapses, preventing manual labeling with potential biases. The authors trained and successfully evaluated ADeS in packed epithelial monolayers and T cells distributed in 3D collagen hydrogels. Moreover, in vivo, training and evaluation were performed on polymorphonucleated leukocytes in lymph nodes and spleen.Strengths:Pulfer A. et colleagues convincingly presented their results, thoroughly evaluated ADeS for potential toxicity assay, and compared its performance with available state-of-the-art tools.Weaknesses:The use of ADeS is still restricted to samples where cells are fluorescently labeled either in the cytoplasm or in the nucleus, which limits its use for in vitro toxicity assays that are performed on primary cells or organoids (e.g., iPSCs-derived systems) that are normally harder to transfect. In conclusion, ADeS will be a useful tool to improve output quality and accelerate the evaluation of assays in several research areas with basic and applied aims.

As addressed in the answer to reviewer one, we primarily focused on fluorescent microscopy, which implies fluorescent labeling of the cells. The application to other imaging platforms was not the scope of our study. However, a model to infer apoptosis within other imaging solutions, e.g. brightfield, could be explored in future analogue studies.